# Rescue of dendritic cells from glycolysis inhibition improves cancer immunotherapy in mice

Sahil Inamdar[1,11], Abhirami P. Suresh[2,11], Joslyn L. Mangal [2], Nathan D. Ng[3], Alison Sundem[1], Christopher Wu[4], Kelly Lintecum [1], Abhirami Thumsi[2], Taravat Khodaei[4], Michelle Halim[1], Nicole Appel[1,5], Madhan Mohan Chandra Sekhar Jaggarapu[1], Arezoo Esrafili[1], Jordan R. Yaron [1], Marion Curtis [6,7] & Abhinav P. Acharya [1,2,4,5,8,9,10] ✉

Inhibition of glycolysis in immune cells and cancer cells diminishes their activity, and thus combining immunotherapies with glycolytic inhibitors is challenging. Herein, a strategy is presented where glycolysis is inhibited in cancer cells using PFK15 (inhibitor of PFKFB3, rate-limiting step in glycolysis), while simultaneously glycolysis and function is rescued in DCs by delivery of fructose-1,6-biphosphate (F16BP, one-step downstream of PFKFB3). To demonstrate the feasibility of this strategy, vaccine formulations are generated using calcium-phosphate chemistry, that incorporate F16BP, poly(IC) as adjuvant, and phosphorylated-TRP2 peptide antigen and tested in challenging and established YUMM1.1 tumours in immunocompetent female mice. Furthermore, to test the versatility of this strategy, adoptive DC therapy is developed with formulations that incorporate F16BP, poly(IC) as adjuvant and mRNA derived from B16F10 cells as antigens in established B16F10 tumours in immunocompetent female mice. F16BP vaccine formulations rescue DCs in vitro and in vivo, significantly improve the survival of mice, and generate cytotoxic T cell (Tc) responses by elevating Tc1 and Tc17 cells within the tumour. Overall, these results demonstrate that rescuing glycolysis of DCs using metabolite-based formulations can be utilized to generate immunotherapy even in the presence of glycolytic inhibitor.

Glycolysis inhibition in cancer cells has been long recognised as a viable strategy to prevent cancer cell growth[1,2]. In fact, cancer cells are known to have a high glucose consumption rate as a metabolism adaptation mechanism called the Warburg effect (or aerobic glycolysis)[3,4], where these cells generate ATP from glycolysis rather than oxidative phosphorylation (OXPHOS) even in the presence of high oxygen supply[5,6]. Thus, several pharmacological approaches have been tested to prevent glycolysis in cancer cells that include drugs such as trametinib, vemurafenib and dacarbazine as well as targeting various glycolytic enzymes and transporters[1,7–9]. Unfortunately, activated immune cells, also rely on glycolysis for their energy needs[10,11]. For example, glycolysis is essential for dendritic cells (DCs) to get activated and generate TNFα, which helps prime T cells against melanoma tumours for recognition and their eventual elimination[10,12]. However, blocking glycolysis in cancer cells while simultaneously allowing glycolysis to occur in DCs is a major challenge in immunometabolism and cancer immunotherapy.

Immunometabolism reprogramming is an emerging and exciting field for generating effective cancer immunotherapy[13,14]. Glycolysis reprogramming in DCs can be performed by overexpressing glycolytic enzymes or glucose receptors, however, these require genetic

manipulation of immune cells, which can be challenging[11,15,16]. Another approach to accelerate glycolysis can be to provide glycolytic metabolites exclusively to DCs, which can then satisfy the energy needs of these cells while blocking the glycolysis of cancer cells. However, these metabolites are small molecules that get consumed or diffuse away after administration in vivo[17,18]. Therefore, there is a need to generate technologies that can provide a sustained release of metabolites to DCs. Notably, DCs are phagocytes, and thus phagocytosable particles that release glycolytic metabolites in a sustained manner can be effective in allowing glycolysis to occur in this cells[19]. Here we show that providing systemic glycolytic inhibitors to block glycolysis in cancer cells while providing metabolite-based phagocytosable particles to antigen-presenting cells, can allow for the generation of DC-based immunotherapy, even in the presence of pharmacological glycolytic inhibitors (Fig. 1a).

## Results

### F16BP microparticles are in phagocytosable range

To demonstrate the strength of such a strategy PFK15 (blocks−6-Phosphofructo-2-Kinase Fructose-2,6-Biphosphatase 3-PFKFB3) was chosen to block glycolysis, as it is one of the rate-limiting steps in glycolysis[20]. Moreover, Fructose-1,6-BiPhosphate (F16BP), which is generated by phosphofructokinase (PFK) that is a step downstream of PFKFB3 was chosen as the metabolite to generate phagocytosable particles[21]. F16BP-based microparticles were generated using calcium-phosphate chemistry. Dynamic light scattering demonstrated that the size of these particles was $2.3 \pm 0.4$ µm (Fig. 1b) and scanning electron micrographs showed that these particles had smooth spherical morphology (Fig. 1c). Using 1H NMR and EDX-mapping it was determined that the F16BP was incorporated within these microparticles (Fig. 1c; S1) and the particles had $2 \pm 0.14$ of Calcium to Phosphorous (Ca:P) ratio (Fig. 1c).

### F16BP microparticles are phagocytosed by DCs

To test if F16BP MPs could release F16BP, release kinetics in phosphate-buffered saline was performed. It was observed that F16BP MPs could release F16BP for 6 h in a sustained manner (Fig. S2). These data demonstrated that the MPs generated contain F16BP, are in the phagocytosable range of DCs and can release F16BP to potentially allow glycolysis to move forward. In addition to F16BP, particles of other control metabolites such as ribulose 5 phosphate (R5P), Phosphoenolpyruvic acid (PEP), and fructose-6 phosphate (F6P) were also generated (Fig. S3). In this study, PBS was chosen to mimic in vivo physiological conditions of phosphates, and the release of F16BP in this medium was determined[22]. Moreover, since activated DCs are expected to survive for 1–3 days in vivo upon phagocytosis, the short-term activity or stability of the F16BP MPs is desirable[23].

To test if these particles can be phagocytosed by DCs, confocal imaging was performed. Specifically, F16BP MPs were generated with FITC intercalated within the particles. Bone marrow-derived DCs (DCs) were then incubated with these particles for 60 min, and stained for actin and nuclei, and fluorescent imaging was performed. Cytochalasin D in the presence of F16BP-FITC MPs were used as control. It was observed that DCs were able to associate with the particles effectively, and the confocal slices in the z-direction demonstrated that the particles were internalised (Fig. S4).

### F16BP microparticles rescue glycolysis in DCs in vitro

Next, the ability of the F16BP MPs to rescue glycolysis in DCs in the presence of glycolytic inhibitor PFK15 was tested using extracellular flux assays. DCs were cultured with either PFK15, F16BP MPs, soluble Fructose-6-Phosphate (F6P-upstream of PFKFB3) or PFK15 + F16BP MPs, or individual components of the F16BP MPs for 2 h, and extracellular acidification rate (ECAR−the rate of glycolysis) measurements

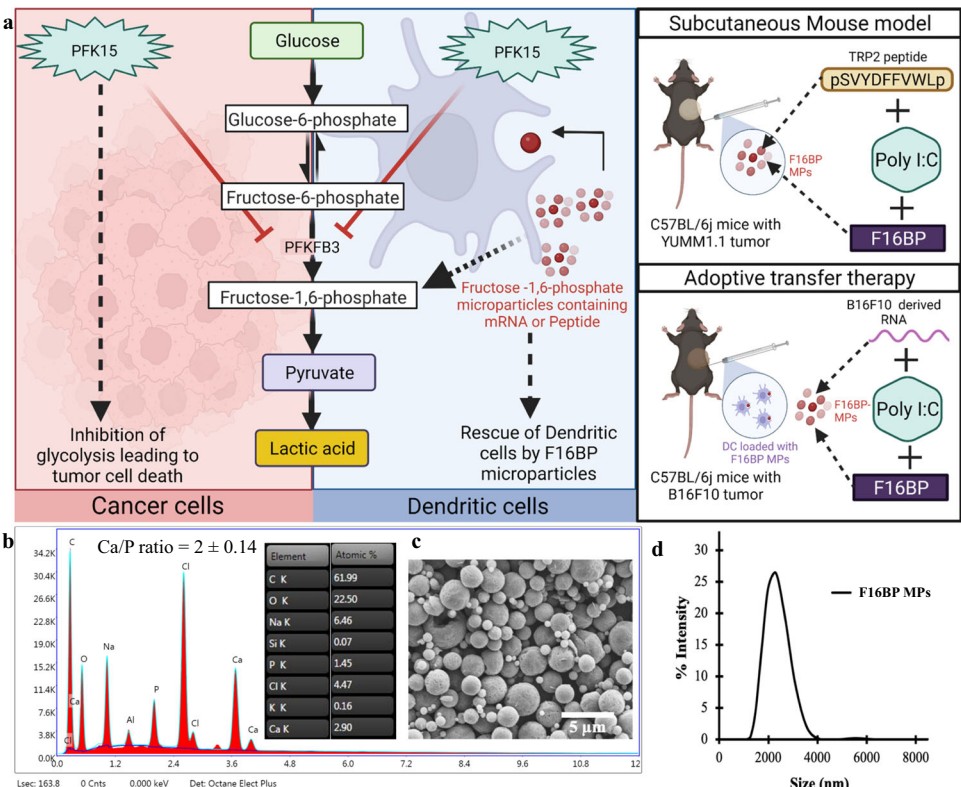

**Fig. 1 | Fructose-1,6-biphosphate (F16BP), a glycolytic metabolite, can be formulated into microparticles (MPs). a** Schematic representation of F16BP MPs rescuing dendritic cells (DCs) from glycolytic inhibition. **b** Scanning electron microscopy-energy dispersive X-ray analysis (SEM-EDX) mapping demonstrating the presence of Ca and P in the microparticles. **c** Scanning electron microscopy image indicating spherical morphology. **d** Dynamic light scattering (DLS) suggests a size of $2200 \pm 300$ nm of F16BP MPs.

were obtained. It was observed that PFK15 brought the glycolysis and glycolytic capacity (y-axis-ECAR) lower than the no-treatment control (Fig. 2a–d; S5). Importantly, glycolysis and glycolytic capacity were significantly higher in F16BP MPs even in the presence of the PFK15, as compared to the PFK15 alone control (Fig. 2a–d). Also, in the presence of poly(IC) (an activating agent for DCs)[24], PFK15 decreased the glycolysis and the glycolytic capacity, and the F16BP MPs were able to rescue this decrease even in the presence of PFK15 (Fig. 2e–h). Overall, these data suggest that the phagocytosable F16BP MPs were able to rescue glycolysis in DCs even in the presence of glycolytic inhibitor PFK15, and thus can be utilised for immunotherapies where the glycolysis pathway of cancer cells is targeted. This is important since under conditions of inflammation, DCs need to perform enhanced levels of glycolysis to support the inflammatory protein production[11,25].

## F16BP microparticles rescue activation of DCs in vitro

To test if F16BP MPs can modulate the function of DCs, mRNA-based and peptide-based vaccine F16BP MPs were formulated. These vaccines were generated by intercalating di-phosphorylated melanoma peptide antigen Tyrosine-related protein 2 (pTRP2), and poly(IC) and mRNA derived from melanoma cells, which contain several phosphate groups in their backbone (Fig. S6). Since pTRP2p, and poly(IC) have phosphate groups, these molecules can be incorporated into the F16BP MPs using the calcium-phosphate chemistry. The amount of pTRp2 and poly(IC) incorporated within the F16BP MPs was determined to be $78 \pm 3.4$, and $50.9 \pm 7.9\,\mu g$, respectively. Next, F16BP MPs intercalated with poly(IC) and pTRP2 were incubated with DCs overnight, and flow cytometry was utilised to test if these particles could activate DCs (Fig. 2i, j). It was observed that the vaccine particles induced significantly higher frequency of MHCII + CD86+ in CD11c+ DCs as compared to the individual component controls of the MPs (Fig. 2i). Additionally, it was observed that PFK15 was able to decrease the activation of DCs (CD80 + CD86+ in CD11c + ) even in the presence of poly(IC), and F16BP MPs were able to rescue the activation of DCs even in the presence of PFK15 (Fig. 2j). Also, calcium ions at the concentration that were added to the DCs as the F16BP MPs, did not lead to changes in activation (MHCII + CD80+ in CD11c + ) profile of DCs (Fig. S5). These data indicate that the F16BP-based vaccine MPs rescue DC activation even in the presence of PFK15, which is important if in vivo cancer vaccine responses need to be generated in the presence of glycolysis inhibition.

To further analyse if DCs treated with F16BP MP formulations modulate T-cell responses, a syngeneic mixed lymphocyte reaction (MLR) was performed. C57BL/6j bone marrow-derived DCs were treated with different conditions (Fig. S7) for 2 h and then cultured with T cells isolated from C57BL/6j mice for 60 h. The cells were then stained against CD4, CD8, CD44, Tbet, RORγT, GATA3, CD25 and Foxp3, and analysed using flow cytometry. It was observed that the F16BP MPs, F16BP(pTRP2), F16BP(poly(IC), F16BP(pTRP2+poly(IC)), PFK15 + F16BP MPs, and PFK15 + F16BP(pTRP2+poly(IC)) all significantly upregulated the frequency of activated Th1, activated Th17, and activated Tc1 cells, while simultaneously downregulating the frequency of Th2, Tregs, and activated Th2 (Fig. S7). Interestingly, it was observed that the treatment of DCs with F16BP MPs led to the biggest changes in T-cell polarisation and activation. This change observed was even in the presence of adjuvant poly(IC) or the antigen pTRP2. Moreover, soluble F16BP and its components added to the DCs induced a significantly lower frequency of activated Th1, Tc1, and Th17 as compared to F16BP MPs in all possible combinations. These data suggest that the presence of particles was important for skewing pro-inflammatory T-cell frequencies in an MLR reaction.

## F16BP vaccines with glycolytic inhibitors generate robust anti-tumour responses

To test if the immunometabolism modulating approach of rescuing glycolysis in the presence of glycolytic inhibitors and generating

cancer vaccine immunotherapies, highly aggressive forms of melanoma mouse models were chosen. Specifically, vaccine MPs were injected subcutaneously in mice containing YUMM1.1 (murine BRAF$^{v600e}$ mutation similar to humans) melanoma tumours, and their ability to reduce tumour growth and modulate innate and adaptive immune responses was tested.

In this melanoma model, $0.75 \times 10^6$ YUMM1.1 cells were injected subcutaneously in C57BL/6j immunocompetent mice, and PFK15 was injected every alternate day for the duration of the study. Moreover, F16BP(pTRP2+poly(IC)) were injected subcutaneously on the same days as the PFK15 injections (Fig. 3a). In vitro it was determined that PFK15 was effective in preventing the proliferation of YUMM1.1 cancer cells (Fig. S8). It was observed that the treatment group of PFK15 + vaccine MPs led to significantly increased survival of mice (day 60 - endpoint) and slower tumour growth as compared to all the different controls (Fig. 3b; S9). This increase in survival in the vaccine MPs was associated with an increase in the DC population and activated DC population in the draining inguinal lymph nodes on day 35, which suggests that the vaccine MPs were able to modulate the innate immune responses (Fig. 3c, d). F16BP was essential in generating anti-tumour responses in mice since without F16BP mice did not survive beyond day 45. Moreover, 3 times the dosage of F16BP (pTPR2+ poly IC) MPs without PFK15 was able to abrogate existing tumours, which were not detected till day 60 (Fig. 3b; S9). Additionally, it was also found that the poly(IC) and pTRP2 needed to be incorporated within the F16BP MPs, and the injections of F16BP MPs with soluble (poly(IC) + pTRP2), did not reduce tumour growth in mice (Fig. S9). F16BP MPs by themselves might accelerate the glycolysis in different cells in mice, as was observed in vitro in DCs. This acceleration of glycolysis then might cause the immune response to be skewed toward anti-tumour responses. However, this control was not tested in mice and is a limitation of this study. Within the tumour, no significant differences were observed in the number of CD4 + T cells of different treatment groups; however, there was a significant increase in the proliferating and activated CD4+ cells in mice treated with vaccine MPs as compared to the other treatment groups (Fig. S10a). Furthermore, no significant differences were observed in the number of T-helper type 1 (Th1) and activated and proliferating Th1 cells; a significant increase in T-helper type 17 (Th17) and activated and proliferating Th17 cells were observed in mice treated with vaccine MPs as compared to other treatment groups (Fig. S10a). It was also observed that when mice were treated with PFK15 and with soluble pTRP2 and soluble poly (IC) without F16BP, it led to increased levels of activated DCs, as compared to a no-treatment control, however, DCs were not modulated in other organs (Fig. S10b). Furthermore, without F16BP, the formulation did not modulate pro-inflammatory T-cell responses as compared to the no-treatment control (Fig. S10c). These data suggest that F16BP was required in the formulation to generate pro-inflammatory T-cell responses. Moreover, within the tumour, there was a significant increase in the number of CD8 + T cells and proliferating and activated CD8 + T cells in mice treated with vaccine MPs as compared to other treatment groups indicating the decrease in tumour growth kinetics in mice treated with vaccine MPs responses (Fig. 3e, f). No significant differences were observed in regulatory T cells in the different treatment groups (Fig. 3g). Additionally, there was a significant increase in the number of Tc1, Tc17 and proliferating and activated Tc1 and Tc17 cells in mice treated with vaccine MPs as compared to other treatment groups (Fig. 3h–k). Also, a significant increase in the Tc1/Treg ratio was observed in mice treated with vaccine MPs as compared to other treatment groups (Fig. 3l). These data demonstrate that the vaccine MPs were able to generate robust pro-inflammatory adaptive immune responses in the tumour.

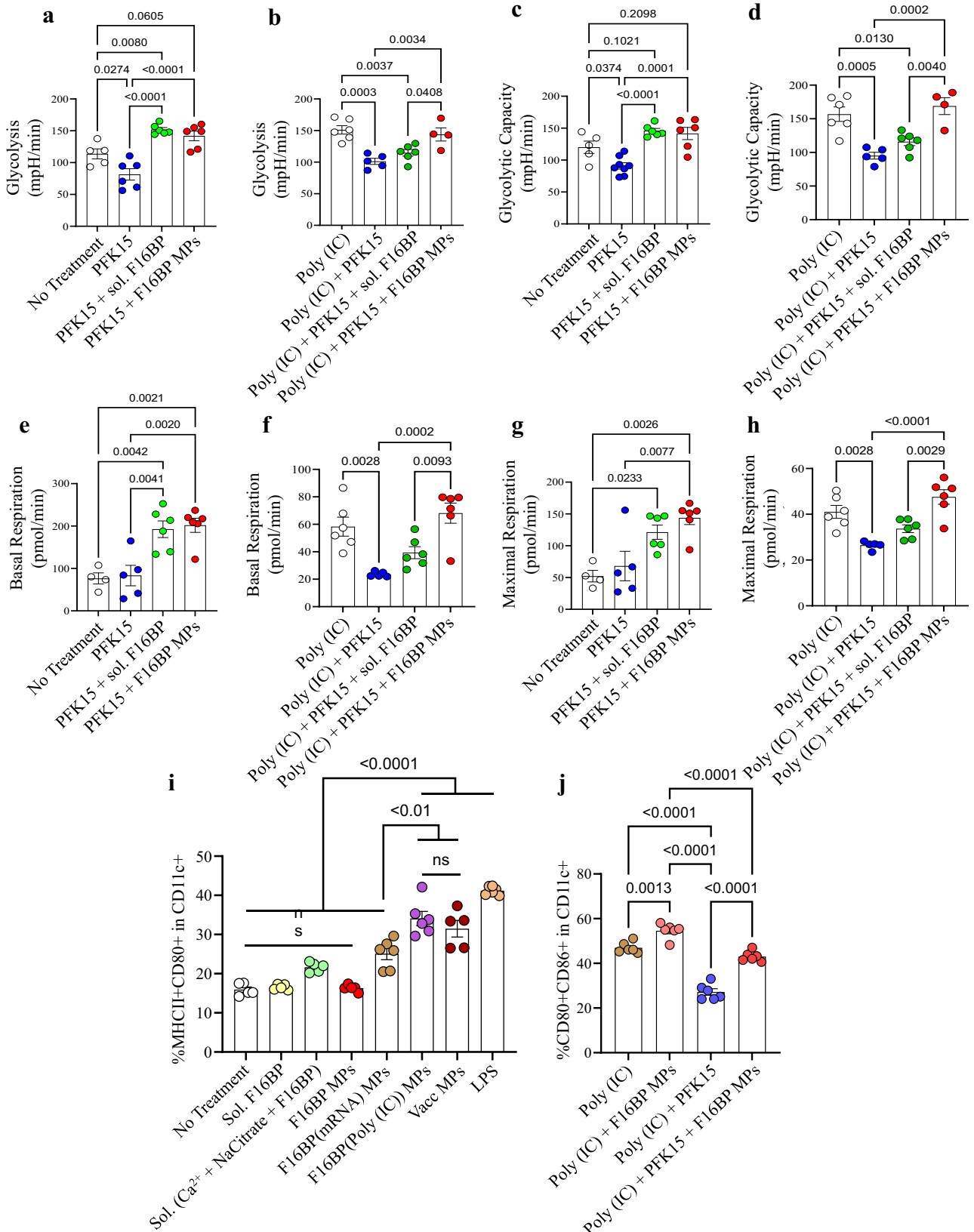

**Fig. 2 | F16BP MPs rescue DC glycolysis and function. a–d** DCs treated with F16BP MPs rescued glycolysis and glycolytic capacity from glycolytic inhibition (PFK15), in vitro (n = 6; One-way ANOVA Tukey's test), **e–h** DCs treated with F16BP MPs accelerate basal and maximal respiration even under glycolytic inhibition (PFK15), in vitro (n = 6; One-way ANOVA Tukey's test). **i** Vaccine particles induced significantly higher frequency of MHCII + CD86+ in CD11c+ DCs as compared to the individual component controls of the MPs (n = 6; One-way ANOVA Tukey's test). **j** F16BP MPs were able to rescue the activation of DCs even in the presence of PFK15 (PFK15 conc. = 25 μM) (n = 6; One-way ANOVA Tukey's test). Data represented as mean ± std error.

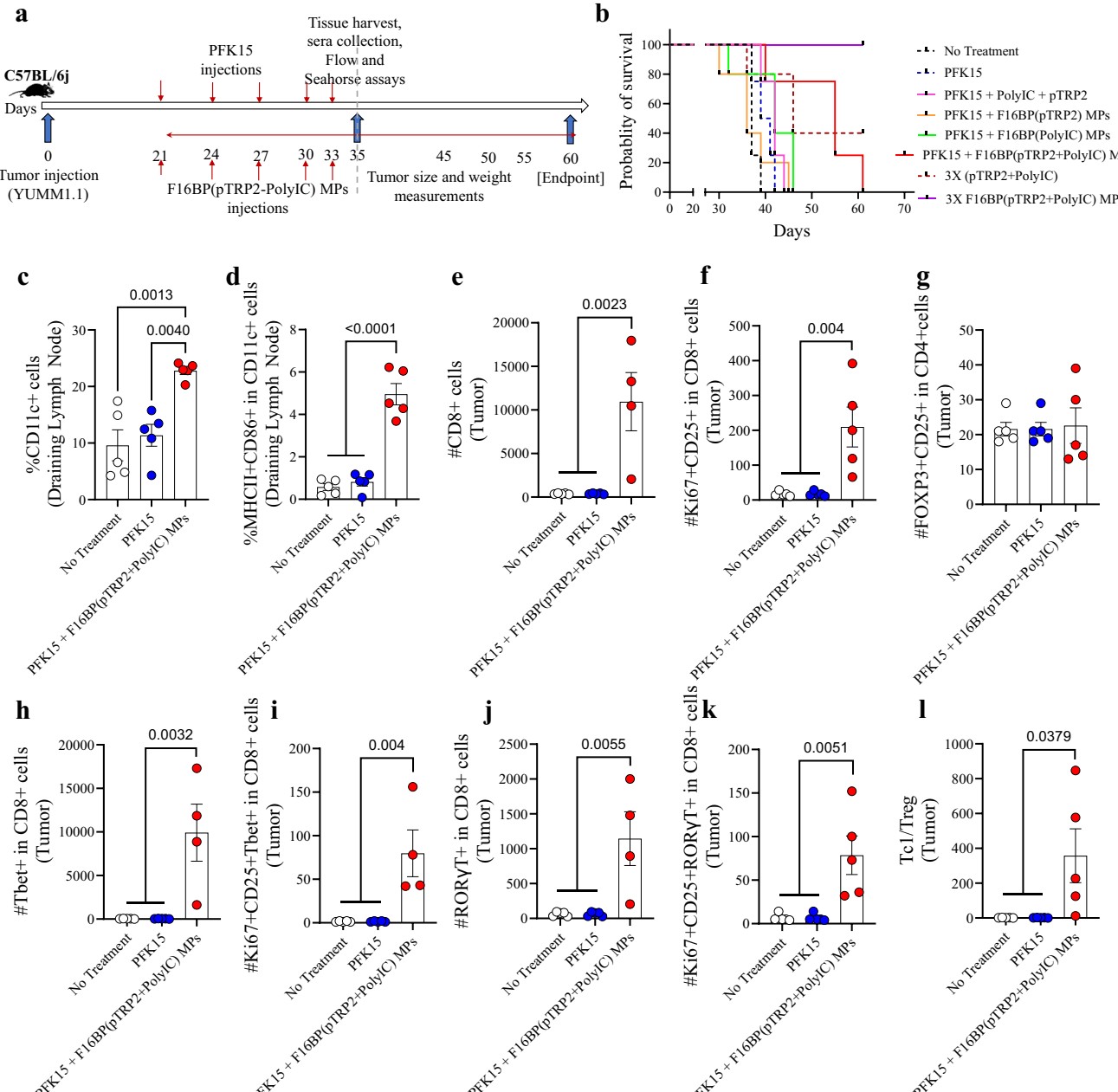

**Fig. 3 | F16BP(pTRP2+PolyIC) MPs promote a pro-inflammatory response against melanoma, in vivo. a** Schematic representation of subcutaneous injection of Vacc MPs, in vivo, **b** Kaplan–Meir curve demonstrating significantly higher survival of mice treated with Vacc MPs, **c, d** Mice treated with Vacc MPs had a significantly higher percentage of the total as well as activated DCs in the draining lymph ($n = 5$; One-way ANOVA Tukey's test), **e, f** Mice treated with Vacc MPs had significantly higher number of Tc and activated and proliferating Tc as compared to other treatment groups ($n = 4$ or 5; One-way ANOVA Tukey's test), **g** No significant differences in the number of Tregs was observed across treatment groups ($n = 5$; One-way ANOVA Tukey's test), **h–k** Mice treated with Vacc MPs had significantly higher number of Tc, Tc1, activated and proliferating Tc1 and Tc17 cells ($n = 4$; One-way ANOVA Tukey's test), **l** Significantly higher Tc1/Treg ratio was observed in mice treated with Vacc MPs as compared to the control groups ($n = 4$; One-way ANOVA Tukey's test). Data represented as mean ± std error.

## F16BP vaccines maintain DC and T-cell metabolic function in vivo

To test if the DCs or T cells maintain their metabolic function after treatment, mice with tumours were euthanized and the cells from tumours, spleen and inguinal LNs were isolated. These cells were then cultured with 2NBDG and flow was utilised to determine the uptake of 2NBDG representing the level of glycolysis. It was observed that in the tumour, gMFI of 2NBDG in CD80+ DCs and macrophages were significantly higher in PFK15 + F16BP(polyIC) MPs as compared to PFK15 only condition, however, these were not significantly different than no-treatment control (Fig. S11a–g). In the spleen, the gMFI of 2NBDG in

DCs, CD80+ DCs, but not in CD206+ DCs were significantly higher in PFK15 + F16BP(polyIC) MPs treated mice as compared to all the controls (Fig. S11h–j). These trends were reversed in macrophages isolated from the spleen of PFK15 + F16BP(polyIC) MPs treated mice as compared to all the controls (Fig. S11k–m). These data suggest that systemically the MPs differentially modulated glycolysis of DCs and macrophages. In the inguinal LNs, gMFI of 2NBDG also was upregulated in activated CD80+ DCs, and CD206+ DCs (Fig. S11n–s). The 2NBDG assay thus demonstrated that the DC and macrophage glycolysis was still maintained in a tumour, spleen and in draining inguinal LNs. A similar study was performed for adaptive T cells to understand

glycolytic plasticity in these cells. Notably, it was observed that CD45- cells isolated from the tumour had significantly lower 2NBDG gMFI in PFK15, PFK15+ soluble F16BP + soluble poly(IC), and PFK15 + F16BP(polyIC) MPs treated mice as compared to the control of no-treatment (Fig. S12a). This data suggest that the MPs or soluble parts of the MPs were not able to substantially modulate the glycolysis of non-immune cells or these cells have a higher level of metabolic plasticity as compared to immune cells. Moreover, gMFI of 2NBDG in T-helper cells in the tumour was not significantly different in PFK15 + F16BP(polyIC) MPs as compared to the no-treatment control, and these two conditions were significantly higher than the other controls (Fig. S12b). The gMFI of 2NBDG in CD8 + T cells in the tumour was not significantly different in PFK15 + F16BP(polyIC) MPs as compared to the no-treatment control (Fig. S12c), but was higher than PFK15+ soluble F16BP + soluble poly(IC) condition. In the spleen, there were no significant differences observed in CD4 + T cells, however, CD8 + T cells had higher 2NBDG gMFI as compared to the controls in PFK15 + F16BP(polyIC) MPs treated mice, but not different than no-treatment control (Fig. S12d, e). Also, in the inguinal lymph nodes, PFK15 + F16BP(polyIC) MPs treated mice had lowered 2NBDG gMFI as compared to the no-treatment control (Fig. S12f, g). The T-cell 2NBDG assay suggests that in the tumour both CD4+ and CD8+ T cells maintain their glycolysis even after ex vivo culture, and thus may support anti-tumour responses in vivo. Overall, these data demonstrated that the vaccine MPs that deliver F16BP and can rescue DCs were able to generate robust immune responses against an aggressive form of melanoma tumours.

### F16BP rescues adoptively transferred DC metabolism in mice

In addition to the subcutaneous vaccine strategy, adoptive transfer of DCs has been tested in clinics for the treatment of prostate cancer[26–28]. However, these strategies have not been very successful in clinics in part due to their low efficacy in survival improvement. To test the versatility of the F16BP-PFK15 system, another aggressive B16F10 melanoma model was chosen and the ability of adoptive transfer of DCs were loaded with the MPs was utilised as a treatment modality (Fig. 4a). In this model, DCs were loaded with F16BP MPs intercalated with mRNA isolated from B16F10 cancer cells, and poly(IC) (Vacc DCs) and adoptively transferred in mice containing B16F10 tumours. This cellular therapy's ability to modulate adaptive immune responses, reduce tumour growth and improve survival was measured.

First, to test if the F16BP MPs (without poly(IC) or mRNA) can rescue DCs in vivo, F16BP MPs were loaded into DCs by incubating these particles with DCs for 2 h. Next, these DCs were adoptively transferred intravenously in mice and PFK15 was injected intraperitoneally. DCs were isolated from the spleen and extracellular flux assays were performed on these cells. It is expected that these DCs isolated from the spleen will be a mixture of both endogenous splenic DCs and adoptively transferred DCs. It was observed that the F16BP MPs were able to rescue glycolysis and glycolytic capacity of DCs as observed by the increased ECAR values in the presence of PFK15 as compared to the controls (Fig. 4b, c). Moreover, F16BP MPs were also able to rescue basal and maximal respiration of DCs in the presence of PFK15, as compared to the controls (Fig. 4d, e). This data demonstrated that these particles can not only rescue glycolysis and mitochondrial respiration in these cells in vitro but also in vivo, and thus might be able to generate functional immunotherapeutic responses.

### Metabolic rescue of DCs generates robust adoptive cell immunotherapy

To test if the metabolic rescue in DCs can be applied to adoptive cell therapy, the adoptive transfer of DC vaccines was tested in mice (Fig. S13). First, in vitro, it was determined that PFK15 was effective in preventing the proliferation of B16F10 cancer cells (Fig. S14). Specifically, $0.75 \times 10^6$ B16F10 cells were injected in C57BL/6j

immunocompetent mice, and PFK15 was injected every alternate day for the duration of the study (Fig. S13). Moreover, ex vivo DCs loaded with F16BP(mRNA+poly(IC)) (Vacc DCs) or DCs loaded with F16BP MPs were adoptively transferred using retro-orbital injections (RO) on days 6 and 19 (Fig. S13). It was observed that the survival of mice receiving Vacc DCs + PFK15 increased dramatically as compared to all the controls, and the tumour grew slower in mice receiving Vacc DCs as compared to the controls (Fig. 5a; S15). This data was further corroborated by tumour weights and images that were obtained at the half-point of the study on day 16 (Fig. 5b, c). These data indicate that after just two injections of Vacc DCs, there was a robust response against the tumour, which then reduced the tumour growth in mice.

Next, to test if the Vacc DCs could modulate the immune responses after their administration, on day 16, the spleen, lymph nodes, and tumours were isolated from different conditions and stained for the activation profile of DCs, and T cells. It was observed that the total number of DCs and activated DCs in tumours for Vacc DCs was significantly higher as compared to all the controls (Fig. 5d, e). Moreover, Vacc DCs also upregulated number of DCs that were MHCI + CD86 + CD11c+ in the tumour as compared to other treatment groups, suggesting the Vacc DCs support the activation of DCs in vivo (Fig. 5f). Also, Vacc DCs also upregulated number of DCs that were MHCII + CD86 + CD11c+ in the tumour as compared to adoptively transferred F16BP DCs, suggesting the Vacc DCs skew toward MHCII associated responses in vivo (Fig. 5g). These data indicate that glycolysis might be needed for potential infiltration of the DCs in the spleen and the tumour, and F16BP MPs provide the ability to DCs for glycolysis to occur.

The adaptive T-cell responses in the tumour were skewed toward pro-inflammatory responses (Fig. 6a–g; S16). It was observed that PFK15 treatment alone significantly decreased the total number of CD8+ T cells in the tumour, and that DCs loaded with F16BP, were able to bring this number equal to the no-treatment control (Fig. 6a). Within this CD8+ T-cell numbers, it was observed that, there was 2–3-fold increase in activated and proliferating (Ki67+CD44+ in CD8+) T cells in the tumour in Vacc DC group as compared to the control groups (Fig. 6b). Furthermore, there was 4-fold significant increase in cytotoxic Tbet+ in CD8+ T cells (Tc1) in the Vacc DCs group as compared to all the other groups (Fig. 6c). Moreover, the number of activated and proliferating Tc1 cells was >10-fold higher in Vacc DCs group as compared to all the other treatment conditions (Fig. 6d). Similarly, the number of Tc17 cells (RORγt+ in CD8+), and activated and proliferating Tc17 (RORγt+Ki67+CD44+ in CD8+), which are emerging as an important pro-inflammatory cell types that induce cancer cell death, were also >10-fold significantly higher than the controls in the tumours (Fig. 6e, f). In addition to the Tc populations, Vacc DCs also led to a decrease in the numbers of T-helper cells (Th), activated and proliferating Th cells, Th1 (Tbet+ in CD4+), Th2 (GATA3+ in CD4+), activated and proliferating Th1 (Tbet+CD44+Ki67+ in CD4+), and activated and proliferating Th2 cells (GATA3+CD44+Ki67+ in CD4+) numbers in the Vacc DC group (Fig. S16). There was a significant decrease in the number of Treg (CD25+Foxp3+ in CD4+) within the tumour in mice treated with Vacc DCs as compared to untreated mice (Fig. S16). Moreover, there was a significant increase in the Th17 (RORγt+ in CD4 + ) and activated and proliferating Th17 cells (RORγt+Ki67+CD44+ in CD4+) within the tumour in the Vacc DC-treated mice as compared to all the other groups (Fig. S16). Furthermore, the ratio of Tc1 to Treg cells was 5–10-fold higher in the tumour of the Vacc DC-treated mice as compared to the controls (Fig. 6g). These data suggest that the Vacc DCs were able to induce a robust adaptive immune response against the tumours, which might be primarily driven by increases in Tc1 and Tc17 cell populations.

Since Vacc DCs are administered systemically, spleen and cervical lymph nodes were also analysed for the changes in innate and adaptive immune responses (Figs. S17–S20). Notably, the overall frequency of

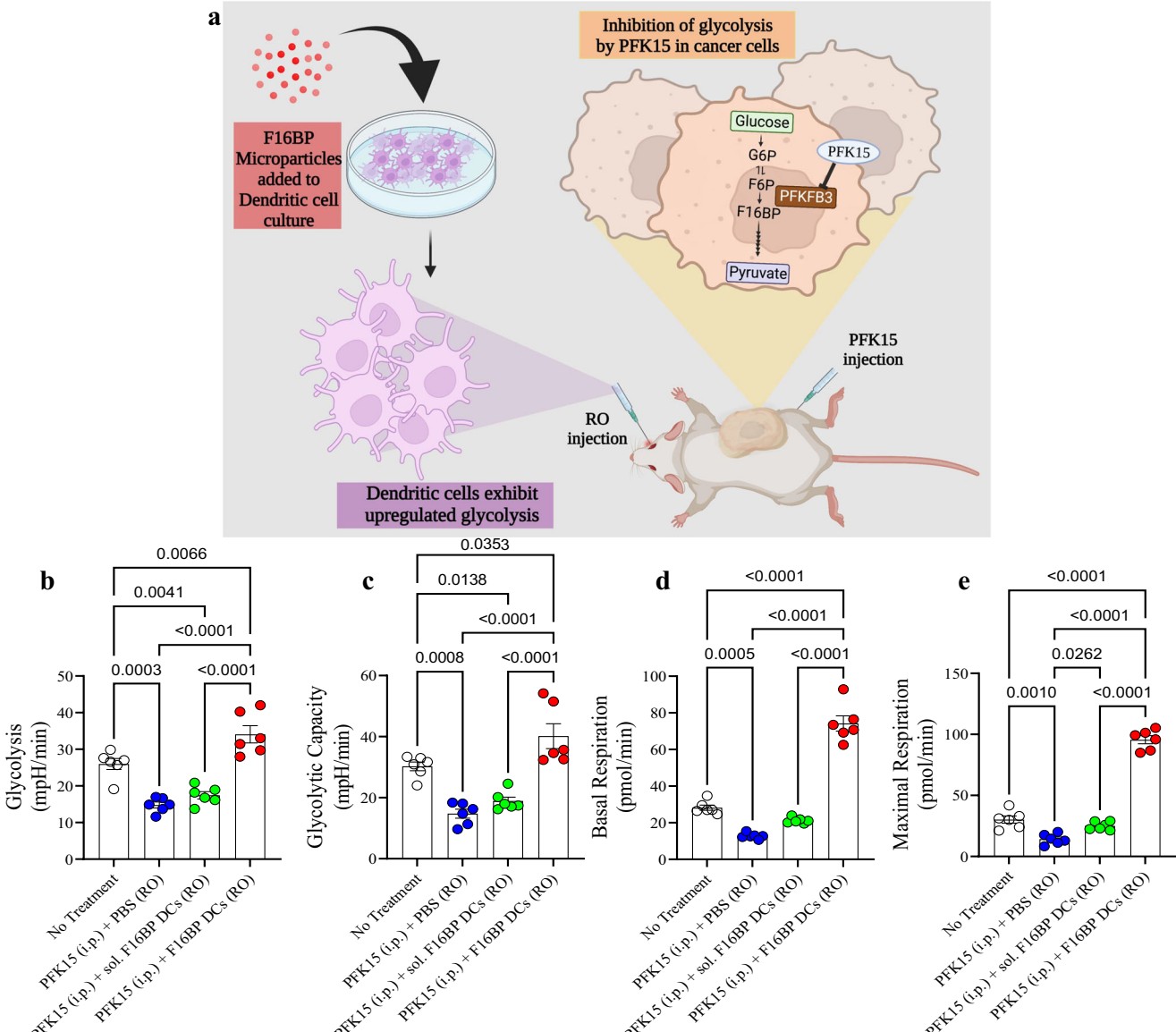

**Fig. 4 | F16BP MPs rescue DCs in adoptive cellular transfer (ACT) therapy model, in vivo. a** Schematic representation of the adoptive cellular therapy model employed. **b, c** Mice injected with adoptively transferred DCs along with F16BP MPs were able to rescue glycolysis and glycolytic capacity from glycolytic inhibition (PFK15), in vivo ($n = 6$; One-way ANOVA Tukey's test), **d, e** Mice injected with adoptively transferred DCs along with F16BP MPs accelerated basal and maximal respiration even under glycolytic inhibition (PFK15), in vivo ($n = 6$; One-way ANOVA Tukey's test). Data represented as mean ± std error.

DCs (CD11c+) and activated DCs (MHCII+CD86+ in CD11c+, MHCI +CD86+ in CD11c+, CD86+ in CD11c+, MHCI+ in CD11c+, MHCII+ in CD11c+) were significantly higher in spleen as compared to the controls; however, this difference was not observed in the cervical lymph nodes (Figs. S17 and S18). These findings suggest that the adoptive transfer of DCs primarily generates immune responses by modulating the innate immune responses in the spleen. Contrary to the tumour, it was observed that in the spleen, there were significant decreases in the Th17 frequency in Vacc DC groups as compared to all the controls (Fig. S19), significant increases in the Th2 frequency as compared to all the controls (Fig. S19), and a significant decrease in Tc17 frequency as compared to PFK15 alone, and PFK15 + F16BP DCs controls (Fig. S20). These findings indicate that although the Vacc DCs were administered systemically, the pro-inflammatory adaptive immune responses were primarily found in the tumour and not in the spleen. Additionally, the Vacc DCs isolated from the spleen of mice also maintained higher ECAR and OCR as compared to the controls (Fig. S21). To test if the F16BP component of the formulation is important for generating

innate and adaptive immune responses DCs loaded with soluble mRNA + soluble poly (IC) were injected in mice retro-orbitally and the innate and adaptive immune responses generated in iLN, spleen and tumours were tested and compared to no-treatment control. It was observed that for increasing pro-inflammatory both DCs and T-cell responses F16BP loaded in DCs was essential, as there were no significant differences observed between no-treatment control and DCs loaded with soluble mRNA + soluble poly (IC) in these organs (Fig. S22).

## Discussion

This study demonstrated that formulations can be utilised to rescue glycolysis of DCs even in the presence of the glycolytic inhibitor. Importantly, this strategy can be utilised to generate vaccine-based immunotherapy in the presence of glycolytic inhibitor chemotherapy. Importantly, it was observed that glycolytic rescue in DCs in two different melanoma model shows that cytotoxic T cells were increased in these treatment groups, potentially due to the rescue of DCs.

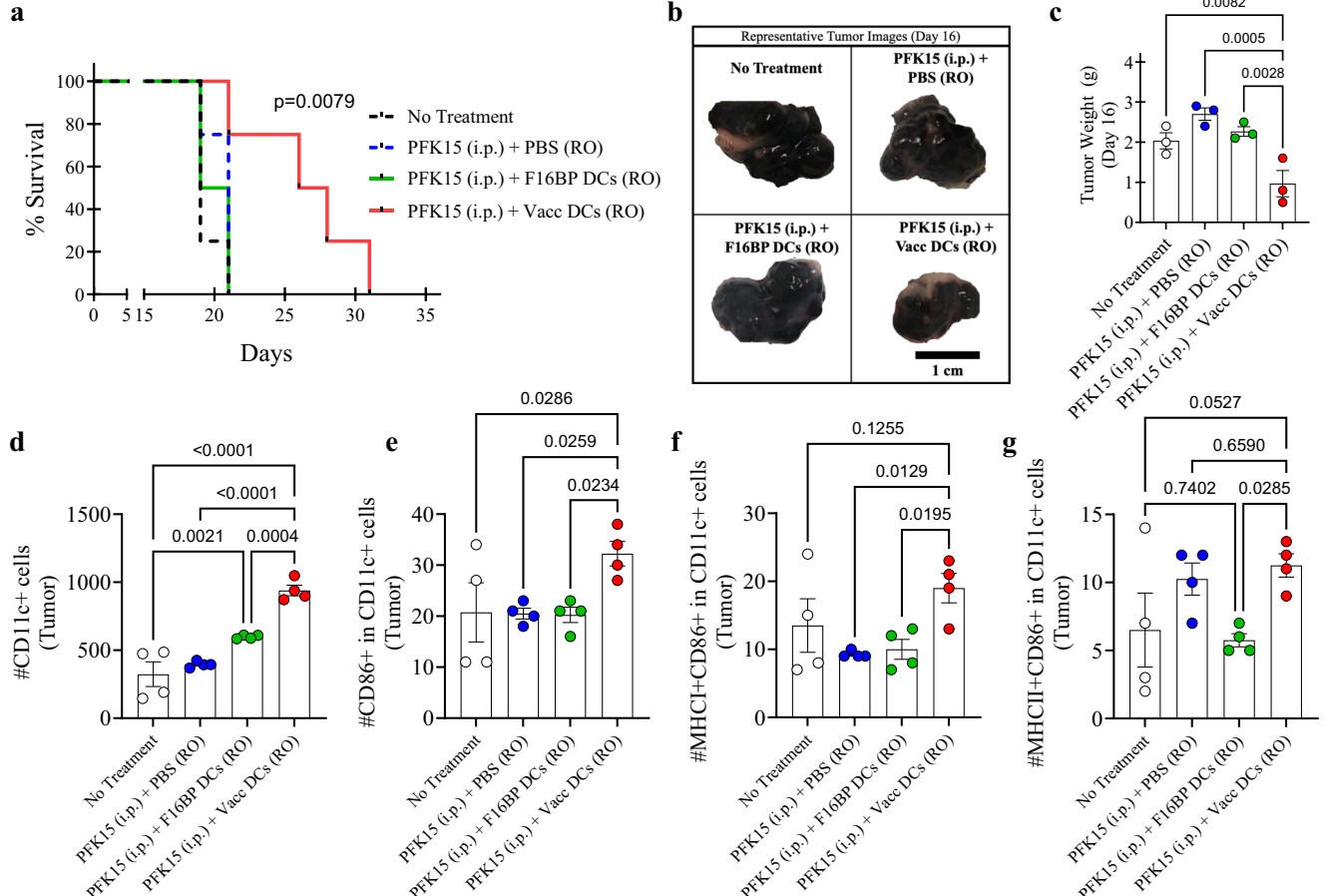

**Fig. 5 | F16BP MPs-based vaccines are compatible with adoptively transferred DCs and improve survival in melanoma. a** Kaplan–Meir curve demonstrating significantly higher survival of mice treated with adoptively transferred Vacc MPs ($n = 10$, $p < 0.001$), **b** Representative tumour images of different treatment groups on day 16. **c** Mice treated with adoptively transferred Vacc DCs had significantly lower tumour weights as compared to other treatment groups, in vivo ($n = 3$; One-way ANOVA Tukey's test). **d, e** Significantly higher total, as well as activated DCs, were observed in mice treated with adoptively transferred Vacc DCs as compared

to other treatment groups, in vivo ($n = 4$; One-way ANOVA Tukey's test). **f** Significantly higher MHCI+ activated DCs were observed in mice treated with adoptively transferred Vacc DCs as compared to other treatment groups, in vivo ($n = 4$; One-way ANOVA Tukey's test). **g** Significantly higher MHCII+ activated DCs were observed in mice treated with adoptively transferred Vacc DCs as compared to F16BP DCs, in vivo ($n = 4$; One-way ANOVA Tukey's test). Data represented as mean ± std error.

This work is important because it is versatile and can be utilised for different nodes in the metabolic pathways of cancer cells such as the TCA cycle, glutaminolysis and mitochondrial respiration. It has been shown that melanoma cells rely on glycolysis[29], which was targeted in this study. Moreover, it has been shown that PFKFB3 is an important target in the glycolytic pathway for retarding cancer cell growth[21,30]. Therefore, clinical trials targeting different types of cancers have been carried out with varying success[31,32]. This limited success can be because different activated immune cells such as DCs, macrophages, and T cells heavily rely on glycolysis for their energy needs, and for long-term cancer remission, it is important to have a robust immune response. Although, in this study we demonstrate that DCs can be rescued from glycolysis inhibition, other immune cells can also be rescued by either targeted delivery of metabolite-based formulation or via adoptive cell therapy. These immune cells were not tested in this manuscript because the scope of this research focused on DCs, and vaccine design. A limitation of this study was that although an increase in survival of mice bearing large tumours prior to the treatment with glycolytic inhibitor and the vaccine F16BP MPs was observed this survival can be further improved. For example, the therapies used in this study can be combined with checkpoint inhibitors and/or IL-2 therapy to further improve the efficacy of survival in mice[33]. It is important to note that combinatorial therapy for melanoma clinical trials with Flt3L, DEC205/NY-ESO-1

fusion protein and poly-ICLC (NCT02129075) have demonstrated immunogenicity and safety. Moreover, clinical trials for low-grade B-cell lymphoma of intratumoral poly-ICLC injections along with radiation (NCT01976585)[34], clinical trials for poly-ICLC and peptide-pulsed autologous DCs in patients with pancreatic cancer have also provided positive outcomes[35]. Furthermore, treatment of mice with poly(IC) loaded adoptively transferred DCs has shown success against melanoma[36,37]. In this study, the effect of multiple injections with adoptively transferred DCs or increased loading of antigen and adjuvant could be considered to completely abrogate the tumours, as has been seen in other studies. These conditions were not tested in this study and is a limitation of this report.

In addition to the metabolite-based formulations, it has been extensively demonstrated that formulations can be developed that can be delivered subcutaneously for developing robust immunotherapy[38–40]. However, these formulations are difficult to combine with chemotherapy, which is the first line of defence in the clinic. Interestingly, in a clinic in addition to glycolytic inhibitors other chemotherapy such as BRAF inhibitors are used, which also lead to a decrease in glycolysis in DCs, and other immune cells. Therefore, a strategy that can improve glycolysis in immune cells will be highly beneficial.

In conclusion, this study demonstrates for the first time that the glycolysis of DCs can be rescued both in vitro and in vivo using a

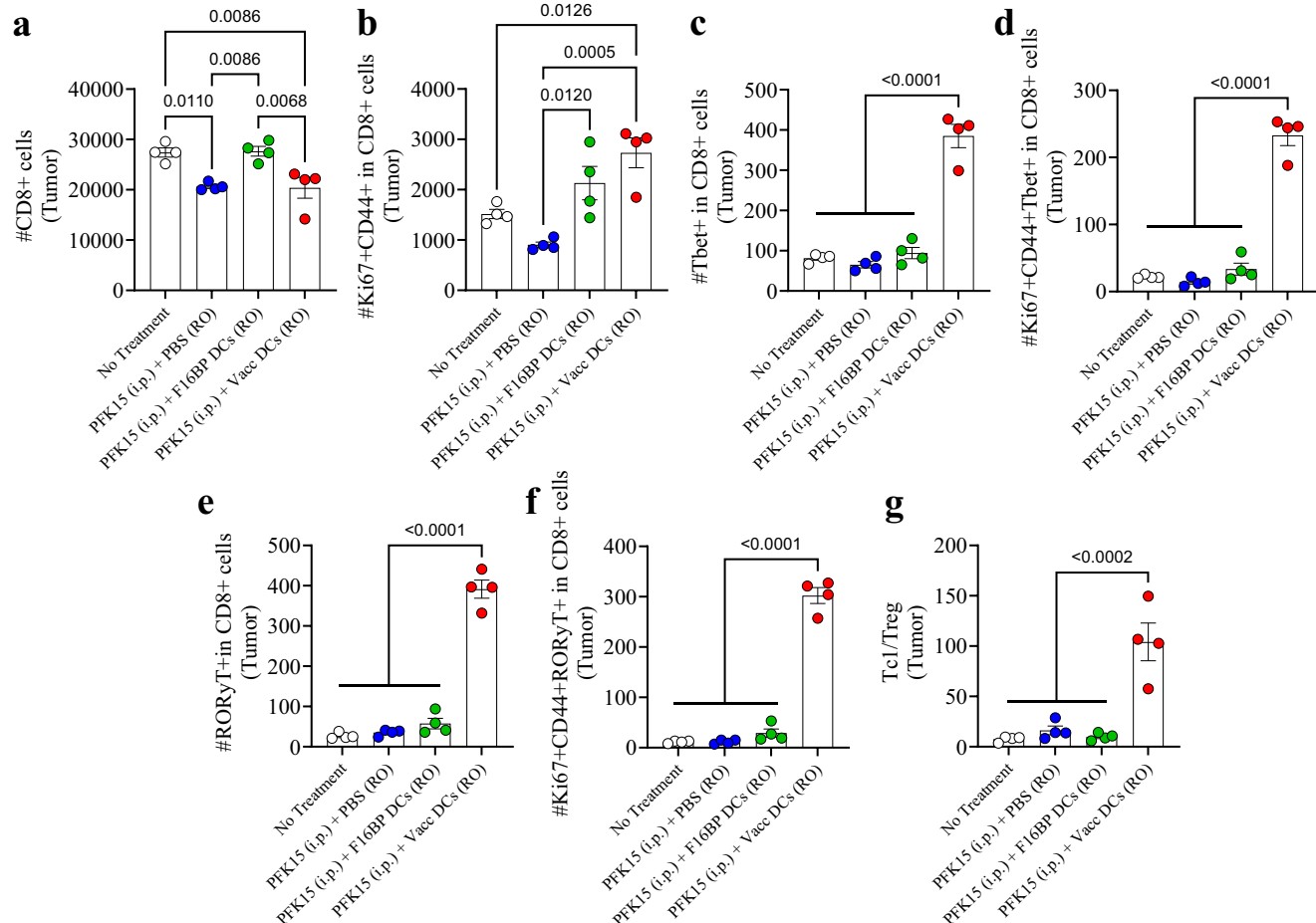

**Fig. 6 | Adoptively transferred F16BP MPs-based vaccines DCs generated robust anti-tumour adaptive immune responses. a, b** Significantly modulation of total (CD8+) as well as activated and proliferating (Ki67+CD44+ in CD8+) cytotoxic T cells were observed in mice treated with adoptively transferred Vacc DCs as compared to other treatment groups, in vivo ($n = 4$; One-way ANOVA Tukey's test). **c–f** Significantly higher total Tc1 (Tbet+ in CD8+), activated and proliferating Tc1 (Tbet+Ki67+CD44+ in CD8+), total Tc17 (RORγT+ in CD8+), activated and proliferating Tc17 (RORγT+Ki67+CD44+ in CD8+) were observed in mice treated with adoptively transferred Vacc DCs as compared to other treatment groups, in vivo ($n = 4$; One-way ANOVA Tukey's test). **g** Significantly higher ratio of cytotoxic to regulatory T cells (Tc1/Treg) was observed in mice treated with adoptively transferred Vacc DCs as compared to other treatment groups, in vivo ($n = 4$; One-way ANOVA Tukey's test). Data represented as mean ± std error.

biomaterial strategy of delivering metabolites downstream of the inhibitory node. This strategy was tested using two different murine melanoma models in immunocompetent mice using two different treatment regimens to generate robust anti-melanoma innate and adaptive immune responses. Using these strategies tumour growth was significantly delayed, which was driven by Tc1 and Tc17-based responses within the tumour. These data demonstrated that metabolism-targeted chemotherapy can be combined with immunotherapies, by actively modulating DC metabolism. However, it should be noted that the strategy of accelerating glycolysis in DCs needs to be further tested in animals for safety before translation. Specifically, the adoptive transfer of immune cells can be an attractive strategy for generating a therapeutic that will need to undergo special cellular therapy consideration through regulatory agency approvals. Nonetheless, this strategy can be generalised to target other metabolic pathways that both cancer cells and immune cells utilise for their function for generating effective cancer immunotherapy.

## Methods
### Microparticle generation
Freshly prepare 0.2 M CaCl$_2$, 0.1 M Na-Citrate, 0.1 M F16BP and 1 N NaOH were used. Initially, 100 μL of nano-pure water (NPW) was added to a 15 mL falcon tube. Next, 1.5 mL of 0.2 M CaCl$_2$, 1 mL of 0.1 M Na-

Citrate and 1 mL of 0.1 M F16BP were added to the falcon tube. Then 200 μL of 1 N NaOH was added dropwise while continuously stirring for 30 s. The mixture was capped and allowed to stir further for 120 s. After 2 min, the solution was centrifuged at 1200 rcf for 5 min. The supernatant was removed, and the particles were washed three times. After the last wash, the resuspended pellet was placed in a tube and incubated at −80 °C. After 3 h, samples were put in the lyophilizer (freeze dryer) for at least 1–2 days. The particles were then used after lyophilization.

To generate MPs incorporating poly (IC) or mRNA + poly (IC), 100 μL of mRNA derived from cancer cells (Qiagen RNA extraction kit) (10 μg/mL in NPW) or 50 μL of mRNA (10 μg/mL in NPW) + 50 μL of poly (IC) (10 μg/mL in NPW) were used instead of 100 μL of NPW. The rest of the steps were followed as stated above to generate the MPs.

### MTT assay
Cell proliferation was determined using an MTT reagent. Specifically, B16F10 and YUMM1.1 cells were cultured in DMEM/F-12 (1:1) with L-glutamine supplemented with 10% fetal bovine serum and 1% penicillin−streptomycin. Briefly, cells were seeded in flat-bottomed 96-well plates (10,000 cells per well) overnight. On the day of the treatment, PFK15 with varying concentrations were added to YUMM1.1 and B16F10 cells. An equal volume of DMEM/F-12 (1:1) was added in the no-

treatment group as a negative control. For positive control (all dead cells), media from wells was aspirated and methanol was added to the wells for 15 min, ensuring the death of all cells in the well, following which methanol was siphoned off and the same amount of media was re-added to the wells. After 48 hrs, 10 μL of the MTT solution was added to all wells and the plates were placed at 37 °C for 3 hrs in the dark. Supernatants from all the wells were aspirated and 50 μL of DMSO: Methanol (1:1) was added to all wells following which the plates were placed in the dark at 37 °C ensuring delicate stirring of the plates. The number of viable cells was determined by measuring absorbance at 570 nm with a reference wavelength of 670 nm using a plate reader (Speedmax M2e, Sunnyvale, CA).

### Particle size determination
Microparticles were imaged and EDX-mapping was performed using scanning electron microscopy (SEM) XL30 Environmental FEG-FEI at Erying Materials Centre at Arizona State University. Dynamic Light Scattering (DLS) was used to determine the average size of the generated particles.

### Dendritic cell isolation and culture
Bone marrow-derived DCs were generated from 6-8 week-old female C57BL/6j mice in compliance with the protocol approved by the Arizona State University (protocol number 19-1688R) using a modified 10-day protocol. Femur and tibia from mice were isolated and the ends of the bones were trimmed; bone marrow was flushed out with 5 mL wash media and made into a homogenous suspension. Red blood cells (RBC) were lysed by centrifuging the suspension and incubating in 3 mL of 1x RBC lysis buffer for 5 min on ice. The cell suspension was centrifuged and washed with 7 mL wash media before re-suspending in DC media DMEM/F-12 with L-glutamine (VWR, Radnor, PA), 10% fetal bovine serum, 1% sodium pyruvate (VWR, Radnor, PA), 1% non-essential amino acids (VWR, Radnor, PA), 1% penicillin–streptomycin (VWR, Radnor, PA) and 20 ng/ml GM-CSF (VWR, Radnor, PA)). The cells were later seeded in tissue culture-treated T-75 flask (Day 0). On day 2, floating cells were collected, centrifuged, and resuspended in fresh media, respectively, and seeded on ultra-low attachment plates for 7 additional days. Media was changed every day until day 9. On day 9, cells from the ultra-low attachment plates were resuspended and $0.1 \times 10^6$ cells/well were seeded on suitable tissue culture plates for the desired experiments for 1 more day (until day 10) before treatment. Cells in the tissue culture plates were used for further experiments/treatment on day 10. The purity, immaturity, and yield of DCs were verified via flow cytometry. DCs were isolated from at least 3 separate mice for each type of experiment.

### Fluorescent microscopy
On day 10, cells were seeded on a glass slide within 24-well plates and were incubated for 24 h at 37 °C. The cells were then treated with fluorescently labelled FITC-F16BP MPs. The nucleus and cytoplasm were stained with DAPI and rhodamine-phalloidin, respectively. Samples were imaged with a fluorescent microscope using a 20x, lens with a numerical aperture of 1.4. DAPI and fluorescently labelled FITC-F16BP MPs were excited with 405 nm and 561 nm lasers, respectively, coupled with appropriate blue and red channel emission detection. Image dimensions were 1024 × 1024 pixels scanned with a digital zoom of 2x. Cells treated with F16BP MPs only, cytochalasin D+FITC-F16BP MPs and untreated cells were used as negative imaging controls to identify the signal of interest.

### Flow cytometry
Flow cytometry (FACS) staining buffer was prepared by generating 0.1% bovine serum albumin (VWR, Radnor, PA), 2 mM $Na_2$EDTA (VWR, Radnor, PA) and 0.01% $NaN_3$ (VWR, Radnor, PA). Live/dead staining was performed using fixable dye eF780 (ThermoFisher Scientific,

Waltham, MA, USA). All antibodies required for staining were purchased and used as is (BD biosciences, Tonbo Biosciences, BioLegend, Thermo Scientific, Invitrogen). Flow cytometry was performed by following the manufacturer's recommendation and guidelines set by ASU flow cytometry core using Attune NXT Flow cytometer (ThermoFisher Scientific, Waltham, MA, USA). Flowjo v10 was utilised to analyse the data obtained from the flow cytometer.

### Extracellular flux assays
Initially, 24 h prior to the experiment, 200 μL/well of distilled water is displayed in the utility cartridge plate (Seahorse mini Fluxpak XFe96, 102601–100, Agilent Technologies) and hydrated overnight, in a 37 °C non-CO2 incubator. On the day of the experiment, the distilled water is replaced by XF Calibrant (100840–100, Agilent Technologies), and cartridge sensors are immersed into the XF Calibrant and incubated for 1 h in a 37 °C non-CO2 incubator.

Oxidation consumption rate (OCR) was measured using Seahorse Extracellular Flux XF-96) analyser (Seahorse Bioscience, North Billerica, MA. Briefly, 200,000 cells/well were seeded in Seahorse XF-96 plates and cultured. Cells were treated with various treatment groups for 2 hrs. After 2 hrs, for OCR, media was changed to unbuffered DMEM containing 2 mM glutamine, 1 mM pyruvate, and 10 mM glucose following sequential injections of oligomycin (2 mM), 7 Carbonyl cyanide-4 (trifluoromethoxy) phenylhydrazone (FCCP) (1 mM), and antimycin/rotenone (1 mM). The OCR after the injection of oligomycin was a measure of ATP-linked respiration and the OCR after the injection of FCCP represented maximal respiratory capacity. Basal respiration was quantified by measuring OCR prior to the injection of oligomycin.

Similarly, for extracellular acidification rate (ECAR), media was changed to unbuffered DMEM containing 2 mM glutamine, 1 mM pyruvate, and 10 mM glucose following sequential injections of glucose, 7 Carbonyl cyanide-4 (trifluoromethoxy) phenylhydrazone (FCCP) (1 mM), and 2-deoxy glucose. The ECAR after the injection of glucose was a measure of glycolysis and the ECAR after the injection of FCCP represented maximal respiratory capacity. Basal respiration was quantified by measuring OCR prior to the injection of oligomycin.

Dendritic cells were isolated from the organs of mice post-euthanasia, using CD11c+ untouched magnetic beads (Miltenyi Biotech), and within 2 h were utilised for the ECAR/OCR studies.

All samples were analysed with 6 technical replicates. Data are analysed through Wave 2.6.1 Software (Agilent Technologies).

### Tumour induction and treatment for s.c. injection mouse model
Female C57BL/6j mice, 6-8 weeks, were obtained from Jackson Laboratory (Bar Harbor, ME). Experiments were performed in compliance with IACUC guidelines of ASU (protocol no. 19-1688R). Mice were housed in cages with 4 mice per cage, light cycles were controlled from 7 a.m. to 7 p.m. and the mice were kept at ambient temperature and humidity. Melanoma cell line, YUMM1.1, was cultured at 37 °C in a 5% $CO_2$ atmosphere in DMEM/F12 with L-glutamine, 10% fetal bovine serum, and 1% penicillin–streptomycin (culture media). For inoculation, cells were removed from flasks using trypsin solution, centrifuged, and resuspended in 5 mL culture media. Trypan blue exclusion was used to determine cell viability. Furthermore, cells were counted and resuspended in sterile PBS to obtain a solution of $7.5 \times 10^6$ cells/1 mL. Finally, mice were s.c. injected with $0.75 \times 10^6$ cells/mouse (100 μL) into the right thigh. All mice were randomised and divided into 10 mice/group before inoculation with tumour cells. Mice were treated on day 6 intraperitoneally with 25 mg/kg PFK15 (unless otherwise mentioned) and 1 mg of microparticles s.c. (0.5 mg on top of the general thigh area on either side) three times a week. Mice weight and tumour growth were measured and recorded every other day. Tumour growth was measured using a digital calliper and calculated as (longest length*narrowest length$^2$)/2. A maximum size of the tumour of 2 cm in any dimension according to the IACUC.

## Tumour Induction and treatment for adoptive transfer therapy mouse model

Female C57BL/6j mice, 6–8 weeks, were obtained from Jackson Laboratory (Bar Harbor, ME). Experiments were performed in compliance with IACUC guidelines of ASU (protocol no. 19-1688R). Mice were housed in cages with 4 mice per cage, light cycles were controlled from 7 a.m. to 7 p.m. and the mice were kept at ambient temperature and humidity. Melanoma cell line, B16F10, was cultured at 37 °C in a 5% $CO_2$ atmosphere in DMEM/F12 with L-glutamine, 10% fetal bovine serum, 1% penicillin–streptomycin (culture media). For inoculation, cells were removed from flasks using trypsin solution, centrifuged and resuspended in 5 mL culture media. Trypan blue exclusion was used to determine cell viability. Mice were injected s.c. with B16F10 melanoma cell line with $7.5 \times 10^5$ cells/per mouse on Day 0.[46] Prior to adoptive transfer, DCs were cultured ex vivo and treated with the appropriate treatment groups. These DCs were from the same strain (C57BL/6J) as the tumour-bearing mice to minimise any inflammation caused due to allogeneic reactions. On day 6, ex vivo cultured and treated DCs were injected retro-orbitally ($2.5 \times 10^6$ cells in 100 μL in PBS). Following injections were carried out once every week tumour reaches the endpoint of 2 cm. Tumour sizes were measured by calliper 3 times per week and expressed as tumour volume, calculated as (longest dimension × (perpendicular dimension)$^2$)/2. A maximum size of the tumour of 2 cm in any dimension according to the IACUC.

In this study, not only survival but also immunological changes between treatment groups were examined. Notably, for immunological studies mice were sacrificed on day 16 (these groups received treatment only once, on day 6). Tumour, spleen, and lymph nodes were harvested post-mortem. The tissues were processed and stained for innate and adaptive immune cells and analysed using flow cytometry. For the survival study, mice were monitored until the tumour reaches the endpoint.

## Statistics

Statistical analysis calculations were carried out using Microsoft Excel and GraphPad Prism software 9.0. For each of the experiments, the statistical experiment was performed separately. $p$-values < 0.05 was considered statistically significant. All data is expressed in the form of mean ± std error unless otherwise specified.

## Reporting summary

Further information on research design is available in the Nature Portfolio Reporting Summary linked to this article.

## Data availability

Source data are provided in this paper.

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

## Acknowledgements

The authors would like to acknowledge the Flow Cytometry Core, the Regenerative Medicine Imaging Facility, the FEI at Erying Materials Centre, the Advanced Light Microscopy Facilities, and the Department of Animal Care and Technologies at Arizona State University. Additionally, the authors would like to thank Jacquelyn Kilbourne, Juliane Dagget-Vondras and Kenneth Lowe, DACT team at Arizona State University for their help with murine experiments. The authors would also like to acknowledge the start-up funds provided by Arizona State University, and NIH R01AR078343 and NIH R01AI155907, NIH 1R01GM144966-01 and NSF award# 2145877 to APA for the completion of this study.

## Author contributions

S.I. designed and performed experiments, analysed data and wrote the manuscript; A.P.S., A.T., M.H., J.L.M., K.L., N.A., M.M.C.S.J., A.E. performed animal experiments and cell culture; T.K., N.D.N. and A.S. assisted in cell culture and material characterisation; C.W. and M.C. assisted with extracellular flux assays; J.Y. performed confocal imaging, A.P.A. designed experiments and wrote the manuscript.

## Competing interests

A.P.A. has some rights reserved for the technology presented. S.I., A.P.S., J.L.M., N.D.N., A.S., C.W., K.L., A.T., T.K., M.H., N.A., M.M.C.S.J., A.E., J.Y., M.C. have no competing interests.

## Additional information

¹Chemical Engineering, School for the Engineering of Matter, Transport, and Energy, Arizona State University, Tempe, AZ 85281, USA. ²Biological Design, Arizona State University, Tempe, AZ 85281, USA. ³Molecular Biosciences and Biotechnology, The College of Liberal Arts and Sciences, Arizona State University, Tempe, AZ 85281, USA. ⁴Department of Biomedical Engineering, School of Biological and Health System Engineering, Arizona State University, Tempe, AZ 85281, USA. ⁵Center for Immunotherapy, Vaccines and Virotherapy, Arizona State University, Tempe, AZ 85281, USA. ⁶Department of Cancer Biology, Mayo Clinic, Scottsdale, AZ 85259 8, USA. ⁷College of Medicine and Science, Mayo Clinic, Scottsdale, AZ 85259, USA. ⁸Materials Science and Engineering, School for the Engineering of Matter, Transport, and Energy, Arizona State University, Tempe, AZ 85281, USA. ⁹Biodesign Center for Biomaterials Innovation and Translation, Arizona State University, Tempe, AZ 85281, USA. ¹⁰Department of Biomedical Engineering, Case Western Reserve University, Cleveland, USA. ¹¹These authors contributed equally: Sahil Inamdar, Abhirami P. Suresh. ✉e-mail: abhi.acharya@asu.edu

