## [Peer Review File · Nature Communications]

REVIEWERS' COMMENTS:

Reviewer #1 (Remarks to the Author):

This manuscript described a glycolysis-centered combination strategy that simultaneously blocked glycolysis in cancer cells and promoted glycolysis in DCs to realize potent cancer immunotherapy in melanoma-bearing mice. The fructose 1,6 biphosphate (F16BP) loaded into the calcium phosphate formulation enabled to rescue DCs with the systemic administration of glycolytic inhibitor PFK15. Benefiting the F16BP-caused DCs rescue, the authors demonstrated that the generated F16BP particles could apply to therapeutic vaccines and adoptive DC therapy to induce robust cytotoxic T cells-mediated anticancer immune response in tumor-bearing mice. The authors provided an interesting treatment regimen that leveraged the different metabolic demands between cancer cells and DCs to activate the anticancer immune response. Although the results of this study are intriguing and may create new strategies to induce successful immunologic therapies, this strategy showed less translational potential, and the animal model is not clinically relevant. Specific questions and comments are listed below.

Major comments

1. Limited translational potential. On the one hand, PFK15, without approval for clinical use, was employed to block glycolysis in cancer cells through i.p. administration. On the other hand, a complex formulation was developed to co-encapsulate F16BP, adjuvant, and antigen. These elements compromise the translation prospect of this strategy.
2. More control groups will make certain conclusions more convincing. Several experiments lack appropriate controls (details as noted in minor comments below).
3. Given the metabolic plasticity in both cancer cells and immune cells, specific experiments and solid data should be supplemented to prove the glycolysis-involved treatment efficacy.
4. Although the metabolism intervention-mediated cancer therapy developed in this manuscript possess elements of novelty, it is not clear that this work is a significant advance over other metabolism and immunometabolism therapies already existing in the literature.

Minor comments:

1. About the release kinetics data (Figure S2), why did the authors choose phosphate-buffered saline as the medium? Have the authors considered the effect of complexation between phosphate in medium and calcium ions in formulation on the drug release? Besides, the F16BP MPs did not show favorable stability. So, how about the stability of this formulation co-loaded with F16BP, adjuvant, and antigen in the following experiments?
2. The authors should provide the data to present the internalization process of particles in DCs.
3. Since calcium ions are recently reported to possess several bioactive abilities, a calcium ion or calcium phosphate control group should be introduced to evaluate the activation of DCs.
4. A scrambled control group should be introduced to eliminate the potential activating effect of nucleotide sequences on DCs.
5. Some results were sloppily organized. For example, there are two Figure S8 in the SI; many n values in the caption of Figure 3 do not match the real sample numbers of the relevant figures; There is no significance analysis in Figure 3g.

Reviewer #2 (Remarks to the Author):

Inamdar and colleagues have performed study in which the therapeutic potential is explored of using an glycolysis PFK inhibitor in tumor cells in vivo while simultaneously rescuing glycolysis in TA-DCs by providing a DC-targeted glycolysis substrate downstream of PFK. While, conceptually this is very interesting and novel avenue to explore that could move de field forward, the manuscript seems premature as the experimental data do often not support the conclusions drawn by the authors, primarily because several key control conditions and cross validation experiments are missing. This significantly limits the impact of the study. Specifically I have the following concerns:

1) The First and most critical issue is the lack of numerous essential controls:

A: Fig 2i – What is effect on this readout with Poly IC MP only? That is a much more relevant control than LPS.

B: Figure 3b - What is response of PFK15+(pTRP2/polyIC)MPs compared to PFK15+F16BP(pTRP2/polyIC)MPs? This is important to know as this will reveal whether the beneficial effect is dependent on glycolysis restoration or just a function of Antigen presence. Likewise, what is effect of F16BP(pTRP2/polyIC)MPs compared to PFK15+F16BP(pTRP2/polyIC)MPs? Is beneficial effect only observed when glycolysis is inhibited?

C: Figure 3c-l – the increased responses observed in the 3rd conditions in these plots could just be a function on the antigen/TLR treatment and could have nothing to do with the F16BP supplementation. Therefore appropriate controls need to be added to these data.

D: Fig5 and 6: again the whole beneficial effect could be simply due to the fact that antigen/tlr is there, rather than this a direct consequence of F16BP supplementation. Therefore a key control is a condition with mrna/tlr mp but w/o F16BP supplementation

2) Using Scenith technology or 2NBDG, it should be shown that glycolysis is indeed impaired/restored in tumor cells and associated DCs in vivo in the experiments as shown in fig 3, to directly demonstrate the treatments sort the same metabolic effects in vivo in DCs as in vitro (fig2)

3) It is unclear why authors decided to switch to a different tumor model for the ACT therapy. This should be explained

4) several of the experimental details are unclear. For instance in Figure 4: How were these MPs generated exactly? What mRNA was used? How much time is there in between DC transfer and metabolic analysis? What DCs were isolated and tested? Only transferred DCs or also endogenous splenic DCs, and how did they discriminate between the two? This is not clear and should be better explained.

5) the experimental groups, particularly for the survival analyses in figure 5, are too small to be able to draw reliable conclusions especially when the aforementioned additional control conditions are added.

6) In all figures with Seahorse and flow data raw seahorse / flow plots should be shown to get an impression of what the bars graphs are based on.

Reviewer #3 (Remarks to the Author):

Inamdar et al. described therapeutic strategy where glycolysis is inhibited in cancer cells using PFK15, while simultaneously glycolysis was rescued in DCs by delivery of F16BP. Vaccine formulations were generated that incorporated F16BP, poly(IC) as adjuvant, and phosphorylated-TRP2 peptide antigen and tested in challenging and established YUMM1.1 tumours in immunocompetent mice. Furthermore, to test the versatility of this strategy, adoptive DC therapy was developed with formulations that incorporated F16BP, poly(IC) as adjuvant and mRNA derived from B16F10 cells as antigens in established B16F10 tumours. F16BP vaccine formulations rescued DCs in vitro and in vivo, significantly improved the survival of mice, and generated cytotoxic T cell (Tc) responses by elevating Tc1 and Tc17 cells within the tumours.

This interesting report described rather novel strategy with potential implications for therapy. Experiments are quite comprehensive. However, one rather big concern surprisingly was not addressed. Survival improvements were clear albeit rather modest. DC vaccine with Poly:IC adjuvant, previously demonstrated rather strong clinical response in mice. However, in this study, this treatment was absent. Authors showed full vaccine formulation only together with F16BP. Thus, it was not clear if this novel treatment had any benefit over previous strategies. Also, authors did not compare therapeutic effect of vaccine formulation without use of PFK15. It would be critical to assess potential therapeutic use of this approach. Overall, these weaknesses did not

make compelling case in support authors conclusions.

Less critical, but probably desirable, would be to directly assess function of DC in stimulation of T cells. As presented, DC functionality can be derived indirectly by changes in T cells in vivo.

RESPONSE TO REVIEWERS' COMMENTS

We greatly appreciate reviewers' comments and now we have given point by point response to each of the reviewers' comments. We believe that this has significantly improved the study.

Reviewer #1 (Remarks to the Author):

This manuscript described a glycolysis-centered combination strategy that simultaneously blocked glycolysis in cancer cells and promoted glycolysis in DCs to realize potent cancer immunotherapy in melanoma-bearing mice. The fructose 1,6 biphosphate (F16BP) loaded into the calcium phosphate formulation enabled to rescue DCs with the systemic administration of glycolytic inhibitor PFK15. Benefiting the F16BP-caused DCs rescue, the authors demonstrated that the generated F16BP particles could apply to therapeutic vaccines and adoptive DC therapy to induce robust cytotoxic T cells-mediated anticancer immune response in tumor-bearing mice. The authors provided an interesting treatment regimen that leveraged the different metabolic demands between cancer cells and DCs to activate the anticancer immune response. Although the results of this study are intriguing and may create new strategies to induce successful immunologic therapies, this strategy showed less translational potential, and the animal model is not clinically relevant. Specific questions and comments are listed below.

Authors – We thank the reviewer for their encouraging words, and we have now addressed the specific concerns and comments below.

Major comments

1. Limited translational potential. On the one hand, PFK15, without approval for clinical use, was employed to block glycolysis in cancer cells through i.p. administration. On the other hand, a complex formulation was developed to co-encapsulate F16BP, adjuvant, and antigen. These elements compromise the translation prospect of this strategy.

Authors – We appreciate the reviewer's comments on the translational potential of this strategy. We agree that PFK15 has not been clinically approved, which somewhat decreases the translational potential of this therapy. We also agree that the formulation generated has three components that can have regulatory hurdles during translation. We also understand that the way discussion is written suggests that this strategy can be clinically translatable. This technology is under option and being commercialized by Immunometablix, LLC (immunometablix.com). This company has done commercialization opportunity for the adoptive transfer strategy and has funding from the U.S. National Science

Foundation for commercialization (NSF SBIR/STTR grant # 2151586). However, to address the reviewer's concern we have now included sentences in the discussion to provide caution to the reader that this technology will need to be further evaluated and iterations on formulations will be required for commercialization. We have now included the following sentences, "However, it should be noted that the strategy of accelerating glycolysis in DCs needs to be further tested in animals for safety before translation. Specifically, adoptive transfer of immune cells can be an attractive strategy for generating a therapeutic that will need to undergo special cellular therapy consideration through regulatory agency approvals".

2. More control groups will make certain conclusions more convincing. Several experiments lack appropriate controls (details as noted in minor comments below).

Authors – We agree with the reviewer that some of the controls need to be done to further support the claims. We have now done several experiments, which are further addressed specifically in the minor comments section.

3. Given the metabolic plasticity in both cancer cells and immune cells, specific experiments and solid data should be supplemented to prove the glycolysis-involved treatment efficacy.

Authors – We thank the reviewer for inquiring about the metabolic state of immune cells and cancer cells after treatment. Figure 4b-e demonstrate that the adoptively transferred DCs remain metabolically fit even in the presence of glycolytic inhibitor in the spleen. However, the metabolic fitness of these cells in the tumor microenvironment was not tested.

Therefore, we have now done new experiments using 2NBDG uptake in T cells, DCs and cancer cells in the tumor microenvironment to understand if the treatment still have the effect on these cells. Moreover, we also performed this test on DCs and T cells isolated from spleen and lymph nodes.

It was observed that PFK15 indeed reduced glycolysis in B16F10 cells in the tumor microenvironment, and the formulations were not able to rescue this glycolysis. Moreover, the formulation of F16BP+poly(IC) was indeed able to maintain the rescue glycolysis in these DCs and T cells in tumor microenvironment, and in a more pronounced manner in the spleen and lymph nodes. This data is now included in the Supplementary Figure S9 and Figure S10; and the results are now included in the result section, which now state that, "To test if the dendritic cells or T cells maintain their metabolic function after treatment, mice with tumours were euthanized and the cells from tumours, spleen and inguinal LNs were isolated. These cells were then cultured with 2NBDG and flow was utilized to

determine the uptake of 2NBDG representing the level of glycolysis. It was observed that in the tumour, gMFI of 2NBDG in CD80+ DCs and macrophages were significantly higher in PFK15+F16BP(polyIC) MPs as compared to PFK15 only condition, however these were no significantly different than no treatment control (**Figure S9a-g**). In spleen, the gMFI of 2NBDG in DCs, CD80+ DCs, but not in CD206+ DCs were significantly higher in PFK15+F16BP(polyIC) MPs treated mice as compared to all the controls (**Figure S9h-j**). These trends were reversed in macrophages isolated from spleen of PFK15+F16BP(polyIC) MPs treated mice as compared to all the controls (**Figure S9k-m**). These data suggest that systemically the MPs differentially modulated glycolysis of DCs and macrophages. In the inguinal LNs, gMFI of 2NBDG also was upregulated in activated CD80+ DCs, and CD206+ DCs (**Figure S9n-s**). The 2NBDG assay thus demonstrated that the DC and macrophage glycolysis was still maintained in tumour, spleen and in draining inguinal LNs. Similar study was performed for adaptive T cells to understand glycolytic plasticity in these cells. Notably, it was observed that CD45- cells isolated from tumour had significantly lower 2NBDG gMFI in PFK15, PFK15+ soluble F16BP + soluble poly(IC), and PFK15+F16BP(polyIC) MPs treated mice as compared to the control of no treatment (**Figure S10a**). This data suggest that the MPs or soluble parts of the MPs were not able to substantially modulate glycolysis of non-immune cells or these cells have higher level of metabolic plasticity as compared to immune cells. Moreover, gMFI of 2NBDG in T helper cells in tumour was not significantly different in PFK15+F16BP(polyIC) MPs as compared to the no treatment control, and these two conditions were significantly higher than the other controls (**Figure S10b**). The gMFI of 2NBDG in CD8+ T cells in the tumour was not significantly different in PFK15+F16BP(polyIC) MPs as compared to the no treatment control (**Figure S10c**), but was higher than PFK15+ soluble F16BP + soluble poly(IC) condition. In spleen, there were no significant differences observed in CD4+ T cells, however, CD8+ T cells had higher 2NBDG gMFI as compared to the controls in PFK15+F16BP(polyIC) MPs treated mice, but not different than no treatment control (**Figure S10d,e**). Also, in the inguinal lymph nodes PFK15+F16BP(polyIC) MPs treated mice had lowered 2NBDG gMFI as compared to the no treatment control (**Figure S10f,g**). The T cell 2NBDG assay suggests that in the tumour both CD4+ and CD8+ T cells maintain their glycolysis even after *ex vivo* culture, and thus may support anti-tumour responses *in vivo*. Overall, these data demonstrated that the vaccine MPs that deliver F16BP and can rescue DCs, were able to generate robust immune responses against aggressive form of melanoma tumours.”.

4. Although the metabolism intervention-mediated cancer therapy developed in this manuscript possess elements of novelty, it is not clear that this work is a significant

advance over other metabolism and immunometabolism therapies already existing in the literature.

Authors – We appreciate the reviewer’s comments and kindly disagree. This the first manuscript to show direct rescue of glycolysis in immune cells from glycolysis inhibition by generating a formulation that directly feeds into glycolytic pathway. To the best of our knowledge we agree with the reviewer that delivery of glycolysis inhibition (e.g. 2-DG inhibitor) has been demonstrated to modulate cancer cell growth in vitro and in vivo.

However, *till date to the best of our knowledge restarting glycolysis in immune cells in the presence of glycolytic inhibitor has not been demonstrated.* To further clarify this idea, we have now included the following sentences, “This strategy of accelerating glycolysis in specific cell types while blocking glycolysis in other cell types is novel and to the best of our knowledge has not been explored to generate cancer immunotherapy.”.

Minor comments:

1. About the release kinetics data (Figure S2), why did the authors choose phosphate-buffered saline as the medium? Have the authors considered the effect of complexation between phosphate in medium and calcium ions in formulation on the drug release? Besides, the F16BP MPs did not show favorable stability. So, how about the stability of this formulation co-loaded with F16BP, adjuvant, and antigen in the following experiments?

Authors – We appreciate the reviewer’s comments about release kinetics. The release kinetics was performed keeping in mind the different ions already present in the body. For example, phosphate groups are present in the serum at 3.4 to 4.5 mg/dl concentrations, which is similar to the 1X PBS that was used for stability and release of phosphates.

Moreover, in this specific study we are focused on targeting dendritic cells, which have a lifespan of approximately 3 days. Therefore, stability of the particles is not a major concern. The DCs are expected to phagocytose, process and display antigens to the T cells within 24 hours.

To further explain these points following sentences have been included in the manuscript, “In this study PBS was chosen to mimic in vivo physiological conditions of phosphates, and the release of F16BP in this medium was determined²¹. Moreover, since activated DCs are expected to survive for 1-3 days in vivo upon phagocytosis, the short term of activity or stability of the F16BP MPs is desirable²².”

2. The authors should provide the data to present the internalization process of particles in DCs.

Authors – We thank the reviewers for suggesting to demonstrate that the particles are indeed internalized by DCs. To test this we performed phagocytosis assay using confocal microscopy with bone marrow derived dendritic cells and Rhodamine loaded F16BP-poly(IC) microparticles. It was determined that the DCs were able to phagocytose the F16BP particles. We have now included confocal slices in the z-direction in Figure S4 bottom panel. Moreover, we have also included a statement in the results section on which now states that, “It was observed that DCs were able to associate with the particles effectively, and the confocal slices in the z-direction demonstrated that the particles were internalized (Figure S4).”

3. Since calcium ions are recently reported to possess several bioactive abilities, a calcium ion or calcium phosphate control group should be introduced to evaluate the activation of DCs.

Authors – We thank the reviewer for their suggestion of adding a calcium ion control group to our study to evaluate the activation of DCs. We have now performed an experiment using flow cytometry to evaluate the effect of calcium ion on DC activation. It was observed that at the concentrations used for this study, calcium ions alone do not upregulate DC activation. This data is now included in Figure S5. Moreover, we have also included a statement in the results section on, “Also calcium ions at the concentration that were added to the DCs as the F16BP MPs, did not lead to changes in activation (MHCII+CD80+ in CD11c+) profile of DCs (Figure S5).”

4. A scrambled control group should be introduced to eliminate the potential activating effect of nucleotide sequences on DCs.

Authors – We thank the reviewer for their suggestion to add a scrambled control group for eliminating the potential activating effect of nucleotide sequences on DCs. We included poly(IC) with the purpose of activating the DCs. Since, the effect of poly(IC) has been well documented and has been utilized for several decades in research, we did not include a scrambled control. Furthermore, the controls were performed with each individual component to test the effect of these on DC activation as depicted in figure 2.

5. Some results were sloppily organized. For example, there are two Figure S8 in the SI; many n values in the caption of Figure 3 do not match the real sample numbers of the relevant figures; There is no significance analysis in Figure 3g.

Authors – We appreciate the reviewer for pointing this out. We have now corrected the figure numbers and checked throughout the manuscript.

Reviewer #2 (Remarks to the Author):

Inamdar and colleagues have performed study in which the therapeutic potential is explored of using an glycolysis PFK inhibitor in tumor cells in vivo while simultaneously rescuing glycolysis in TA-DCs by providing a DC-targeted glycolysis substrate downstream of PFK. While, conceptually this is very interesting and novel avenue to explore that could move the field forward, the manuscript seems premature as the experimental data do often not support the conclusions drawn by the authors, primarily because several key control conditions and cross validation experiments are missing. This significantly limits the impact of the study. Specifically, I have the following concerns:

1) The First and most critical issue is the lack of numerous essential controls:

A: Fig 2i – What is effect on this readout with Poly IC MP only? That is a much more relevant control than LPS.

Authors – We thank the reviewer for pointing out that in figure 2i, poly IC is a better control than LPS, and we agree. Therefore, poly IC is now included in Figure 2i.

B: Figure 3b - What is response of PFK15+(pTRP2/polyIC)MPs compared to PFK15+F16BP(pTRP2/polyIC)MPs? This is important to know as this will reveal whether the beneficial effect is dependent on glycolysis restoration or just a function of Antigen presence. Likewise, what is effect of F16BP(pTRP2/polyIC)MPs compared to PFK15+F16BP(pTRP2/polyIC)MPs? Is beneficial effect only observed when glycolysis is inhibited?

Authors – We greatly appreciate the reviewer's question about the role of PFK15, pTRP2 and poly (IC) in combination. We performed new experiments where YUMM1.1 tumors were induced in mice and treated with PFK15 + pTRP2 + poly IC (this formulation does not lead to MPs, and remains soluble in phosphate buffered saline), or PFK15+F16BP(pTRP2+poly IC) MPs or pTRP2 + polyIC. In each of these conditions, tumors grew in mice and were not significantly different than no treatment control. These data suggest that the presence of F16BP is essential, and PFK15 was also needed to reduce the tumour growth in mice.

Moreover, we also tested 3 times the dosage of F16BP (pTRP2+ poly IC) MPs without PFK15 in mice, which did not let tumours grow in mice, and were undetected at the end of the study of 50 days. Notably, 3 times the dosage of (pTRP2+ poly IC) in mice, without F16BP allowed for tumours to grow in mice, and were detectable in all mice at the end of the study at 50 days. This data has

now been included in supplementary figure S9. We have also included following sentences in the manuscript, which now state that, “F16BP was essential in generating anti-tumour responses in mice, since without F16BP mice did not survive beyond day 45. Moreover, 3 times the dosage of F16BP (pTPR2+ poly IC) MPs without PFK15 was able to abrogate existing tumours, which were not detected till day 50 (Supplementary Figure S9).”

C: Figure 3c-l – the increased responses observed in the 3rd conditions in these plots could just be a function on the antigen/TLR treatment and could have nothing to do with the F16BP supplementation. Therefore appropriate controls need to be added to these data.

Authors – We agree with the reviewer, and therefore have performed additional controls for figures 3c-l with soluble pTRP2 + poly(IC) without the F16BP component. It was observed that the both the innate and adaptive immune responses generated were not significantly different than the no treatment control. These data have now been included in Figure S10b, and Figure S10c, and the results now state that, “It was also observed that when mice were treated with PFK15 and with soluble pTRP2 and soluble poly (IC) without F16BP, it led to increased levels of activated DCs, as compared to no treatment control, however, DCs were not modulated in other organs (Figure S10b). Furthermore, without F16BP, the formulation did not modulate pro-inflammatory T cell responses as compared to the no treatment control (Figure S10c). These data suggest that F16BP was required in the formulation to generate pro-inflammatory T cell responses.”

D: Fig5 and 6: again the whole beneficial effect could be simply due to the fact that antigen/tlr is there, rather than this a direct consequence of F16BP supplementation. Therefore a key control is a condition with mrna/tlr mp but w/o F16BP supplementation

Authors – We again thank the reviewer for this insightful observation. We have now performed new experiments for figure 5 and figure 6, with the conditions of soluble pTRP2 + poly(IC) without the F16BP component and tested for survival and immune responses. Similar to the figure 3, it was observed that the survival was not improved as compared to the no treatment control. Moreover, the activated DC responses and the adaptive CD8 and CD4 T cell responses were also not significantly different than the no treatment control. These data suggest that the F16BP component was important in generating a robust immune response. These data are now included in the figure SX, and the results now state that, “To test if the F16BP component of the formulation is important for generating innate and adaptive immune responses DCs loaded with soluble mRNA + soluble poly

(IC) were injected in mice retro-orbitally and the innate and adaptive immune responses generated in iLN, spleen and tumours were tested and compared to no treatment control. It was observed that for increasing pro-inflammatory both DCs and T cell responses F16BP loaded in DCs was essential, as there was no significant differences observed between no treatment control and DCs loaded with soluble mRNA + soluble poly (IC) in these organs (Figure S22).”

2) Using Scenith technology or 2NBDG, it should be shown that glycolysis is indeed impaired/restored in tumor cells and associated DCs in vivo in the experiments as shown in fig 3, to directly demonstrate the treatments sort the same metabolic effects in vivo in DCs as in vitro (fig2).

Authors – We thank the reviewer for inquiring about the metabolic state of immune and cancer cells after treatment. Figure 4b-e demonstrate that the adoptively transferred DCs remain metabolically fit even in the presence of glycolytic inhibitor in the spleen. However, the metabolic fitness of these cells in the tumour microenvironment was not tested.

Therefore, we have now done new experiments using 2NBDG uptake in T cells, DCs and cancer cells in the tumor microenvironment to understand if the treatment still have the effect on these cells. Moreover, we also performed this test on DCs and T cells isolated from spleen and lymph nodes.

It was observed that PFK15 indeed reduced glycolysis in B16F10 cells in the tumor microenvironment, and the formulations were not able to rescue this glycolysis. Moreover, the formulation of F16BP+poly(IC) was indeed able to maintain the rescue glycolysis in these DCs and T cells in tumor microenvironment, and in a more pronounced manner in the spleen and lymph nodes. This data is now included in the Supplementary Figure SX; and the results are now included in the result section, which now state that, “To test if the dendritic cells or T cells maintain their metabolic function after treatment, mice with tumours were euthanized and the cells from tumours, spleen and inguinal LNs were isolated. These cells were then cultured with 2NBDG and flow was utilized to determine the uptake of 2NBDG representing the level of glycolysis. It was observed that in the tumour, gMFI of 2NBDG in CD80+ DCs and macrophages were significantly higher in PFK15+F16BP(polyIC) MPs as compared to PFK15 only condition, however these were no significantly different than no treatment control (Figure S9a-g). In spleen, the gMFI of 2NBDG in DCs, CD80+ DCs, but not in CD206+ DCs were significantly higher in PFK15+F16BP(polyIC) MPs treated mice as compared to all the controls (Figure S9h-j). These trends were reversed in macrophages isolated from spleen of PFK15+F16BP(polyIC) MPs treated mice as compared to all the controls (Figure S9k-m). These data suggest that systemically the MPs differentially modulated glycolysis of DCs and macrophages. In the inguinal LNs, gMFI of 2NBDG also was upregulated in activated CD80+ DCs, and CD206+ DCs (Figure S9n-s). The 2NBDG assay thus demonstrated that the DC and macrophage glycolysis was still maintained in tumour, spleen and in draining inguinal LNs. Similar study was performed for adaptive T cells to understand glycolytic plasticity in these cells. Notably, it was observed that CD45- cells

isolated from tumour had significantly lower 2NBDG gMFI in PFK15, PFK15+ soluble F16BP + soluble poly(IC), and PFK15+F16BP(polyIC) MPs treated mice as compared to the control of no treatment (**Figure S10a**). This data suggest that the MPs or soluble parts of the MPs were not able to substantially modulate glycolysis of non-immune cells or these cells have higher level of metabolic plasticity as compared to immune cells. Moreover, gMFI of 2NBDG in T helper cells in tumour was not significantly different in PFK15+F16BP(polyIC) MPs as compared to the no treatment control, and these two conditions were significantly higher than the other controls (**Figure S10b**). The gMFI of 2NBDG in CD8+ T cells in the tumour was not significantly different in PFK15+F16BP(polyIC) MPs as compared to the no treatment control (**Figure S10c**), but was higher than PFK15+ soluble F16BP + soluble poly(IC) condition. In spleen, there were no significant differences observed in CD4+ T cells, however, CD8+ T cells had higher 2NBDG gMFI as compared to the controls in PFK15+F16BP(polyIC) MPs treated mice, but not different than no treatment control (**Figure S10d,e**). Also, in the inguinal lymph nodes PFK15+F16BP(polyIC) MPs treated mice had lowered 2NBDG gMFI as compared to the no treatment control (**Figure S10f,g**). The T cell 2NBDG assay suggests that in the tumour both CD4+ and CD8+ T cells maintain their glycolysis even after *ex vivo* culture, and thus may support anti-tumour responses *in vivo*. Overall, these data demonstrated that the vaccine MPs that deliver F16BP and can rescue DCs, were able to generate robust immune responses against aggressive form of melanoma tumours.”.

3) It is unclear why authors decided to switch to a different tumor model for the ACT therapy. This should be explained

Authors – We appreciate the reviewer’s comments to better explain why another animal model was chosen. We have now included the following sentences in the manuscript to better clarify this study, which now state that, “In addition to the subcutaneous vaccine strategy, adoptive transfer of DCs has been tested in clinic for treatment of prostate cancer²³⁻²⁵. However, these strategies have not been very successful in clinic in part due to their low efficacy in survival improvement. To test the versatility of the F16BP-PFK15 system, another aggressive B16F10 melanoma model was chosen and the ability of adoptive transfer of DCs was loaded with the MPs was utilized as a treatment modality (Figure 4a).”

4) several of the experimental details are unclear.

Authors – We understand the reviewer’s queries and have detailed them below -

For instance in Figure 4: How were these MPs generated exactly? What mRNA was used?.

Authors – These microparticles were generated using the same protocol as figure 1, however, in addition to adding poly (IC), we added mRNA isolated from the B16F10 during the synthesis of these microparticles. Since both these molecules are RNA based, they were chelated in the MPs. To further provide information, we have now included the following sentences in the methods section, “To generate

MPs incorporating poly (IC) or mRNA + poly (IC), 100 µL of mRNA derived from cancer cells (Qiagen RNA extraction kit) (10 µg/mL in NPW) or 50 µL of mRNA (10 µg/mL in NPW) + 50 µL of poly (IC) (10 µg/mL in NPW) were used instead of 100 µL of NPW. Rest of the steps were followed as stated above to generate the MPs.”

We appreciate the reviewer for catching this omission.

How much time is there in between DC transfer and metabolic analysis? What DCs were isolated and tested?

Authors – The DCs were isolated from mice, which requires euthanasia, isolation of DCs using magnetic separation. These cells were utilized within 2 hours of mice sacrifice. We have now added the following sentence for further clarification, “Dendritic cells were isolated from organs of mice post-euthanasia, using CD11c+ untouched magnetic beads (Miltenyi Biotech), and within 2 hours were utilized for the ECAR/OCR studies.”

Only transferred DCs or also endogenous splenic DCs, and how did they discriminate between the two? This is not clear and should be better explained. –

Authors – We greatly appreciate the reviewer’s question about if we discriminated between the endogenous and adoptively transferred DCs. We have now included the following sentences in the manuscript to clarify this point for the readers, which now state that, “It is expected that these DCs isolated from spleen will be a mixture of both endogenous splenic DCs and adoptively transferred DCs.”

5) the experimental groups, particularly for the survival analyses in figure 5, are too small to be able to draw reliable conclusions especially when the aforementioned additional control conditions are added.

Authors – We thank the reviewer for this comment. We have now repeated the experiment shown in figure 5a, and included another control of soluble mRNA + soluble poly IC in the survival study. It was observed that the initial claims of PFK15 + Vaccine DCs prolonging survival were maintained after these experiments. New figure for Figure 5a was generated and is now shown in the manuscript.

6) In all figures with Seahorse and flow data raw seahorse / flow plots should be shown to get an impression of what the bars graphs are based on.

Authors – We thank the reviewer for suggesting to include seahorse plots and flow plots. We have now included seahorse plots and flow cytometry schema for innate and adaptive immune responses in the supplementary figure S5, and figure S21.

Reviewer #3 (Remarks to the Author):

Inamdar et al. described therapeutic strategy where glycolysis is inhibited in cancer cells using PFK15, while simultaneously glycolysis was rescued in DCs by delivery of F16BP. Vaccine formulations were generated that incorporated F16BP, poly(IC) as adjuvant, and phosphorylated-TRP2 peptide antigen and tested in challenging and established YUMM1.1 tumours in immunocompetent mice. Furthermore, to test the versatility of this strategy, adoptive DC therapy was developed with formulations that incorporated F16BP, poly(IC) as adjuvant and mRNA derived from B16F10 cells as antigens in established B16F10 tumours. F16BP vaccine formulations rescued DCs in vitro and in vivo, significantly improved the survival of mice, and generated cytotoxic T cell (Tc) responses by elevating Tc1 and Tc17 cells within the tumours.

This interesting report described rather novel strategy with potential implications for therapy. Experiments are quite comprehensive.

Authors – We truly appreciate the kind words from the reviewer.

However, one rather big concern surprisingly was not addressed. Survival improvements were clear albeit rather modest. DC vaccine with Poly:IC adjuvant, previously demonstrated rather strong clinical response in mice. However, in this study, this treatment was absent.

Authors – We agree with the reviewer that increased length of survival are desirable in these experiments. We also agree that the DC vaccines with poly (IC) adjuvant have previously been shown to abrogate the tumours. Since, these studies aimed at understanding the rescue of glycolysis in the presence of glycolytic inhibitor the poly(IC) concentrations utilized were below that used in the literature. For example, poly(IC) concentrations utilized are typically 10-20 ug/mL, whereas in this study the concentrations utilized with the MPs were at 1 ug/mL in the MPs. However, we do agree with the reviewer that a stronger clinical response could be achieved if the concentration of poly (IC) was to be increased. To address this, we have now included the following sentences in the manuscript, “A limitation of this study was that although an increase in survival of mice bearing large tumours prior to the treatment with glycolytic inhibitor and the vaccine F16BP MPs was observed but this survival can be further improved. For example, the therapies used in this study can be combined with checkpoint inhibitor and/or IL-2 therapy to further improve the efficacy of survival in mice.³⁰ ”

Authors showed full vaccine formulation only together with F16BP. Thus, it was not clear if this novel treatment had any benefit over previous strategies. Also, authors did not compare therapeutic effect of vaccine formulation without use of PFK15. It would be critical to assess potential therapeutic use of this approach. Overall, these weaknesses did not make compelling case in support authors conclusions.

Authors – We agree with the reviewer that further controls are needed to test if F16BP acts in improving the immune responses. Therefore, we have now performed several new experiments, with controls of soluble poly (IC), soluble TRP2 (antigen), and also treatments without PFK15. It was observed that these formulations did not significantly improve survival of mice as compared to the no treatment. Moreover, both the innate and adaptive immune responses arising from these controls were also not significantly different than the control. These data are now included in figures SX, SY and SZ. Moreover, we have also included the results in the manuscript, which now state that, “”

Less critical, but probably desirable, would be to directly assess function of DC in stimulation of T cells. As presented, DC functionality can be derived indirectly by changes in T cells in vivo.

Authors – We appreciate the reviewer’s comments on understanding the effect of treated DCs on T cell responses directly. Therefore, we have now performed new experiments performing syngeneic mixed lymphocyte reactions. Interestingly, it was observed that the F16BP MPs (all different combinations with pTRP2 and poly (IC)) led to differentially increase in activated Th1 and Th17 frequency and decreased regulatory T cell and Th2 frequency. These data demonstrated that the availability of formulation in a particulate format might be essential for generating adaptive T cell responses. These data is now included in the figure SX, and the results are included in the manuscript, which now state that, “To further analyse if DCs treated with F16BP MP formulations modulate T cell responses, a syngeneic mixed lymphocyte reaction was performed. Bone marrow derived DCs and T cells were isolated from C57BL/6j mice and cultured for 60 hours and T cells were stained against CD4, CD8, CD44, Tbet, ROR γ T, GATA3, CD25 and Foxp3 and analysed using flow cytometry. It was observed that the F16BP MPs, F16BP(pTRP2), F16BP(poly(IC), F16BP(pTRP2+poly(IC)), PFK15+F16BP MPs, and PFK15 + F16BP(pTRP2+poly(IC)) all significantly upregulated the frequency of activated Th1, activated Th17, and activatedTc1 cells, while simultaneously downregulating the frequency of Th2, Tregs, and activated Th2 (Figure S7).”

REVIEWERS' COMMENTS:

Reviewer #1 (Remarks to the Author):

The authors have addressed my previous comments.

Reviewer #2 (Remarks to the Author):

Dr Acharya and colleagues have submitted a revised version of their manuscript in various of my concerns and comments have been addressed by performing additional experiments and providing textual clarifications. This has clearly improved the quality of the manuscript. However, some my concerns have not yet been addressed sufficiently. This relates to the following:

- 1) The authors have performed a number of additional control experiments that help to better define whether the reason for a better clinical response is due to F16BP supplementation. However, some of these newly added controls still do not allow one to conclude this. For instance, PFK1 + TRP2 + polyIC is used as additional control for comparison with PFK1 + F16BP(TRP2 + polyIC) MPs. However, the former does not only lack F16BP but also has a different formulation (reagents in solution vs in MPs), so here one cannot tell whether the difference in response is driven by F16BP, the MP formulation or both. The authors either need to add PFK1 + (TRP2 + polyIC) MPs or clearly state in the discussion that one of the limitations of the study is that they formally cannot discriminate between the effects of MP formation and F16BP.
- 2) The flow cytometry gating steps as shown in Fig S5 seem off. Gating in cells, singlets and Mphs/DC populations looks weird.

Minor:

- There are 2 figures S21.
- Title could be improved and be made more concise. For instance: Rescue of dendritic cells from glycolysis inhibition improves cancer immunotherapy in mice

Reviewer #3 (Remarks to the Author):

I was generally positive about the manuscript. However, several issues raised questions. Authors provided response, but I either did not understand it or found it unsatisfactory.

I commented that one rather big concern was that survival improvements were clear albeit rather modest. DC vaccine with Poly:IC adjuvant, previously demonstrated rather strong clinical response in mice. However, in this study, this treatment was absent.

Authors responded that Poly:IC indeed may cause strong response as was shown in the literature. They suggested that their concentrations were lower and provided revision in the text to address this. I am not sure I found this response satisfactory. The main premise of the study is improvement of DC based treatment with targeting glycolysis. So, any conclusion that proposed treatment is potentially beneficial needs to be supported by the side by side evaluation of the data. What if the difference would be very minor even at lower Poly:IC concentrations? Would it undermine the conclusion of the study?

My second comment was regarding controls. Authors seems to add those controls. However, they referred to Figures SX, SY, and SZ that I did not find in the provided materials. Authors stated: "we have included the results in the manuscript which now state that ". ?!!! It would be good if authors provided information that would allow me to assess their response.

I suggested to evaluate the function of DCs. Authors performed additional experiments but the results are confusing. Figure S7 showed that DC without treatment did not activate T cells at all. It would be interesting to know why some basic activation of T cells by DC was not detected. Also, in this figure many conditions were performed and all those conditions showed the same results. The

message from this figure was not clear.

RESPONSE TO REVIEWERS' COMMENTS

We greatly appreciate reviewers' comments and now we have given point by point response to each of the reviewers' comments. We believe that this has significantly improved the study.

Reviewers' comments:

Reviewer #1 (Remarks to the Author):

The authors have addressed my previous comments.

Authors: We appreciate the response.

Reviewer #2 (Remarks to the Author):

Dr Acharya and colleagues have submitted a revised version of their manuscript in various of my concerns and comments have been addressed by performing additional experiments and providing textual clarifications. This has clearly improved the quality of the manuscript. However, some my concerns have not yet been addressed sufficiently. This relates to the following:

1) The authors have performed a number of additional control experiments that help to better define whether the reason for a better clinical response is due to F16BP supplementation. However, some of these newly added controls still do not allow one to conclude this. For instance, PFK1 + TRP2 + polyIC is used as additional control for comparison with PFK1 + F16BP(TRP2 + polyIC) MPs. However, the former does not only lack F16BP but also has a different formulation (reagents in solution vs in MPs), so here one cannot tell whether the difference in response is driven by F16BP, the MP formulation or both. The authors either need to add PFK1 + (TRP2 + polyIC) MPs or clearly state in the discussion that one of the limitations of the study is that they formally cannot discriminate between the effects of MP formulation and F16BP.

Authors: We appreciate the reviewer's comment on investigating the importance of F16BP MPs in driving the tumour responses. We had carried out a study where F16BP MPs were injected in conjunction with soluble poly (IC) and pTRP2. The difference between the formulation that reduces tumour growth and this new formulation is that the poly(IC) and pTRP2 was not incorporated within the MPs. This formulation thus was designed to test whether F16BP MPs just in the presence of these soluble agents will lead to tumour responses. We found that at day 60, this new formulation did not lead to tumour reduction. We have now included the following sentence in the manuscript, "Additionally, it was also found that the poly(IC) and pTRP2 needed to be incorporated within the F16BP MPs, and the injections of F16BP MPs with soluble (poly(IC) + pTRP2), did not reduce tumour growth in mice (Figure S9)."

However, we also appreciate the reviewer's concern that soluble F16BP or F16BP MPs by itself might have a response which was not tested in this study, and is a limitation of this study.

Therefore, we have now included the following sentences in the discussion, which now state that, “F16BP MPs by themselves might accelerate the glycolysis in different cells in mice, as was observed *in vitro* in DCs. This acceleration of glycolysis then might cause the immune response to be skewed toward anti-tumour responses. However, this control was not tested in mice and is a limitation of this study.”

2) The flow cytometry gating steps as shown in Fig S5 seem off. Gating in cells, singlets and Mphs/DC populations looks weird.

Authors: We appreciate the reviewer’s query about the gating schema. The representative analyses were shown from the cells isolated from tumours. Macrophages were considered as CD11b+CD11c+ and DCs were considered as CD11b-CD11c+ for these analyses. We have now included a gating schema of spleen, to demonstrate the different populations, which is representative of the analyses.

Minor:

- There are 2 figures S21.

Authors: We greatly appreciate Reviewer for catching this mistake. We have now corrected this and relabeled the figures as S21 and S22.

- Title could be improved and be made more concise. For instance: Rescue of dendritic cells from glycolysis inhibition improves cancer immunotherapy in mice

Authors: We agree with the reviewer and have now modified the title of the manuscript to, “Rescue of dendritic cells from glycolysis inhibition improves cancer immunotherapy in mice”

Reviewer #3 (Remarks to the Author):

I was generally positive about the manuscript. However, several issues raised questions. Authors provided response, but I either did not understand it or found it unsatisfactory.

Authors: We greatly appreciate that the reviewer found the manuscript generally positive, and are terribly sorry for the oversight of not being clearer in explaining the results. We have now made some effort to explain our results in a bit more detail. We hope the reviewer find these answers satisfactory.

I commented that one rather big concern was that survival improvements were clear albeit rather modest. DC vaccine with Poly:IC adjuvant, previously demonstrated rather strong clinical response in mice. However, in this study, this treatment was absent.

Authors: We thank the reviewer for this comment, and we completely agree with the reviewer that a formulation that completely abrogates the tumour will need to be investigated. We did perform new experiments with subcutaneous vaccination with 3 times the dose of F16BP (polyIC + pTRP2) MPs, which resulted in no tumours observed at the end point of the study. We believe that a similar higher dose in the adoptive cellular therapy might lead to a 100% survival in mice. Therefore, we have now included the following sentences in the manuscript, "It is important to note that combinatorial therapy for melanoma clinical trials with Flt3L, DEC205/NY-ESO-1 fusion protein and poly-ICLC (NCT02129075) have demonstrated immunogenicity and safety. Moreover, clinical trial for low grade B-cell lymphoma of intratumoral poly-ICLC injections along with radiation (NCT01976585)²⁹, clinical trial for poly-ICLC and peptide-pulsed autologous dendritic cells in patients with pancreatic cancer have also provided positive outcomes³⁰. Furthermore, treatment of mice with poly(IC) loaded adoptively transferred DCs has shown success against melanoma^{31,32}. In this study the effect of multiple injections with adoptively transferred DCs or increased loading of antigen and adjuvant could be considered to completely abrogate the tumours, as has been seen in other studies. These conditions were not tested in this study and is a limitation of this report."

Authors responded that Poly:IC indeed may cause strong response as was shown in the literature. They suggested that their concentrations were lower and provided revision in the text to address this. I am not sure I found this response satisfactory. The main premise of the study is improvement of DC based treatment with targeting glycolysis. So, any conclusion that proposed treatment is potentially beneficial needs to be supported by the side by side evaluation of the data. What if the difference would be very minor even at lower Poly:IC concentrations? Would it undermine the conclusion of the study?

Authors: We appreciate the reviewer's comments about level of poly (IC) in treatment of melanoma tumours. We have performed new experiments where we increased the dosage of F16BP (poly IC + pTRP2) MPs 3 fold and tested to see if that changes the tumour growth rate. We also included the controls of soluble poly IC and pTRP2 at 3 fold levels. We then tested to see how these two conditions fared in comparison to other conditions when injected subcutaneously in mice bearing YUMM1.1 tumours. It was observed that by day 60, 3 fold F16BP(poly IC + pTRP2) MPs formulation did not show any tumours and 40% of the mice with soluble formulations did not have tumours. These data suggested that the F16BP which accelerate glycolysis maybe required for prevention of tumour growth. These data support the earlier conclusion of the study that rescue from glycolysis is important for generation of immunotherapy. We have included this data in Figure 3b. We have also included the following sentences, "F16BP was essential in generating anti-tumour responses in mice, since without F16BP mice did not survive beyond day 45. Moreover, 3 times the dosage of F16BP (pTRP2+ poly IC) MPs without PFK15 was able to abrogate existing tumours, which were not detected till day 60 (Figure 3b, S9). Additionally, it was also found that the poly(IC) and pTRP2 needed to be

incorporated within the F16BP MPs, and the injections of F16BP MPs with soluble (poly(IC) + pTRP2), did not reduce tumour growth in mice (Figure S9)."

My second comment was regarding controls. Authors seems to add those controls. However, they referred to Figures SX, SY, and SZ that I did not find in the provided materials. Authors stated:

"we have included the results in the manuscript which now state that ". ?!!! It would be good if authors provided information that would allow me to assess their response.

Authors: We are so sorry to have made this mistake and forgot to attach the responses in the last response to reviewers.

Original comment was, "Authors showed full vaccine formulation only together with F16BP. Thus, it was not clear if this novel treatment had any benefit over previous strategies. Also, authors did not compare therapeutic effect of vaccine formulation without use of PFK15. It would be critical to assess potential therapeutic use of this approach. Overall, these weaknesses did not make compelling case in support authors conclusions."

Following was the original response –

We agree with the reviewer that further controls are needed to test if F16BP acts in improving the immune responses. Therefore, we have now performed several new experiments, with controls of soluble poly (IC), soluble TRP2 (antigen), and also treatments without PFK15. It was observed that these formulations did not significantly improve survival of mice. Moreover, both the innate and adaptive immune responses arising from these controls were also not significantly different than the control.

We greatly appreciate the reviewer's question about the role of PFK15. We performed new experiments where YUMM1.1 tumors were induced in mice and treated with F16BP (poly IC + pTRP2) without PFK15, or PFK15 + soluble pTRP2 + soluble poly (IC) without F16BP control. In each of these conditions, tumors grew in mice and were not significantly different than no treatment control. These data suggest that the presence of F16BP is essential, and PFK15 was also needed to reduce the tumour growth in mice.

Moreover, we also tested 3 times the dosage of F16BP (pTPR2+ poly IC) MPs without PFK15 mice, which did not let tumours grow in mice, and were undetected at the end of the study of 60 days. Notably, 3 times the dosage of (pTPR2+ poly IC) in mice, without F16BP allowed for tumours to grow in mice, and were detectable in all mice at the end of the study at 50 days. This data has now been included in supplementary figure S9. We have also included following sentences in the manuscript, which now state that, "F16BP was essential in generating anti-tumour responses in mice, since without F16BP mice did not survive beyond day 45. Moreover, 3 times the dosage of F16BP (pTPR2+ poly IC) MPs without PFK15 was able to abrogate existing tumours, which were not detected till day 50 (Supplementary Figure S9)."

We also performed additional controls for figures 3c-I with soluble pTRP2 + poly(IC) without the F16BP component, where immunological responses were measured. It was observed that the both the innate and adaptive immune responses generated were not significantly different than the no treatment control. These data have now been included in

Figure S10b, and Figure S10c, and the results now state that, “It was also observed that when mice were treated with PFK15 and with soluble pTRP2 and soluble poly (IC) without F16BP, it led to increased levels of activated DCs, as compared to no treatment control, however, DCs were not modulated in other organs (Figure S10b). Furthermore, without F16BP, the formulation did not modulate pro-inflammatory T cell responses as compared to the no treatment control (Figure S10c). These data suggest that F16BP was required in the formulation to generate pro-inflammatory T cell responses.”

We also performed new experiments for figure 5 and figure 6, with the conditions of soluble pTRP2 + poly(IC) without the F16BP component and tested for survival and immune responses. It was observed that the survival was not improved as compared to the no treatment control. Moreover, the activated DC responses and the adaptive CD8 and CD4 T cell responses were also not significantly different than the no treatment control. These data suggest that the F16BP component was important in generating a robust immune response. These data are now included in the figure SX, and the results now state that, “To test if the F16BP component of the formulation is important for generating innate and adaptive immune responses DCs loaded with soluble mRNA + soluble poly (IC) were injected in mice retro-orbitally and the innate and adaptive immune responses generated in iLN, spleen and tumours were tested and compared to no treatment control. It was observed that for increasing pro-inflammatory both DCs and T cell responses F16BP loaded in DCs was essential, as there was no significant differences observed between no treatment control and DCs loaded with soluble mRNA + soluble poly (IC) in these organs (Figure S22).”

I suggested to evaluate the function of DCs. Authors performed additional experiments but the results are confusing. Figure S7 showed that DC without treatment did not activate T cells at all. It would be interesting to know why some basic activation of T cells by DC was not detected. Also, in this figure many conditions were performed, and all those conditions showed the same results. The message from this figure was not clear.

Authors: This comment is very intriguing to us as well. Since this is a syngeneic mixed lymphocyte reaction (DCs and T cells obtained from the same strain of mice), the activation of T cells in the presence of DCs was not seen for the given time and at the ratio of DC:T cells, but we do find some activation in no treatment, which was even higher in some of the treatment (this data was intriguing and unexpected to us). We regularly see robust activation and proliferation of T cells in allogenic mixed lymphocyte reaction.

However, we do agree that as originally presented the message from this figure was not very clear. It was observed that the presence of F16BP MPs was important in skewing the T cell responses toward pro-inflammatory T cell responses. Accordingly, we have included the following sentences, “Interestingly, it was observed that the treatment of DCs with F16BP MPs led to the biggest changes in T cell polarization and activation. This change observed was even in the presence of adjuvant poly(IC) or the antigen pTRP2. Moreover, soluble F16BP and its components added to the DCs induced significantly lower frequency of activated Th1, Tc1, and

Th17 as compared to F16BP MPs in all possible combinations. These data suggest that the presence of particles was important for skewing pro-inflammatory T cell frequencies in an MLR reaction. “

REVIEWERS' COMMENTS

Reviewer #2 (Remarks to the Author):

The authors have addressed my concerns satisfactorily. I have no further comments and like to congratulate them with this elegant study

Reviewer #3 (Remarks to the Author):

[none]